# Recent Developments in Cassava (*Manihot esculenta*) Based Biocomposites and Their Potential Industrial Applications: A Comprehensive Review

**DOI:** 10.3390/ma15196992

**Published:** 2022-10-09

**Authors:** Walid Abotbina, S. M. Sapuan, R. A. Ilyas, M. T. H. Sultan, M. F. M. Alkbir, S. Sulaiman, M. M. Harussani, Emin Bayraktar

**Affiliations:** 1Advanced Engineering Materials and Composites Research Centre, Department of Mechanical and Manufacturing Engineering, Universiti Putra Malaysia, Serdang 43400, Selangor, Malaysia; 2Sustainable Waste Management Research Group (SWAM), School of Chemical and Energy Engineering, Faculty of Engineering, Universiti Teknologi Malaysia, Johor Bahru 81310, Johor, Malaysia; 3Centre for Advanced Composite Materials (CACM), Universiti Teknologi Malaysia, Johor Bahru 81310, Johor, Malaysia; 4Laboratory of Biocomposite Technology, Institute of Tropical Forest and Forest Products, Universiti Putra Malaysia, Serdang 43400, Selangor, Malaysia; 5Department of Aerospace Engineering, Universiti Putra Malaysia, Serdang 43400, Selangor, Malaysia; 6Advanced Facilities Engineering Technology Research Cluster, Malaysian Institute of Industrial Technology (MITEC), University Kuala Lumpur, Persiaran Sinaran Ilmu, Bandar Seri Alam, Masai 81750, Johor, Malaysia; 7Facilities Maintenance Engineering Section, Malaysian Institute of Industrial Technology (MITEC), Universitiy Kuala Lumpur, Johor Bahru 81750, Johor, Malaysia; 8Energy Science and Engineering, Department of Transdisciplinary Science and Engineering, School of Environment and Society, Tokyo Institute of Technology, Meguro 152-8552, Tokyo, Japan; 9School of Mechanical and Manufacturing Engineering, ISAE-SUPMECA Institute of Mechanics of Paris, 93400 Saint-Ouen, France

**Keywords:** cassava, natural fibers, cellulose, nanocellulose, biopolymers, biobased materials, biocomposites, blend polymer

## Abstract

**Highlights:**

**Abstract:**

The rapid use of petroleum resources coupled with increased awareness of global environmental problems associated with the use of petroleum-based plastics is a major driving force in the acceptance of natural fibers and biopolymers as green materials. Because of their environmentally friendly and sustainable nature, natural fibers and biopolymers have gained significant attention from scientists and industries. Cassava (*Manihot esculenta*) is a plant that has various purposes for use. It is the primary source of food in many countries and is also used in the production of biocomposites, biopolymers, and biofibers. Starch from cassava can be plasticized, reinforced with fibers, or blended with other polymers to strengthen their properties. Besides that, it is currently used as a raw material for bioethanol and renewable energy production. This comprehensive review paper explains the latest developments in bioethanol compounds from cassava and gives a detailed report on macro and nano-sized cassava fibers and starch, and their fabrication as blend polymers, biocomposites, and hybrid composites. The review also highlights the potential utilization of cassava fibers and biopolymers for industrial applications such as food, bioenergy, packaging, automotive, and others.

## 1. Introduction: Natural Fiber and Starch Biopolymer

In recent decades, agro-industrial waste such as husk and cob, arrowroot bagasse, apple pomace, palm sugar, sugar beet pulp, wheat bagasse, sugar cane bagasse, coffee husk/pulp, rice bran/straw, and mango bagasse, as well as cassava, has increased significantly in line with the increase in the world’s population [1,2,3]. Agricultural industrial wastes have gained huge attention due to environmental and health concerns, despite their low energy consumption and low-cost manufacturing, low density, high specific strength and modulus, relatively good performance, renewable nature, biodegradability, wide availability, renewability, and abundance in nature [4]. These agricultural industrial wastes can be converted into biofiber and biopolymer and can be utilized for short shelf-life applications such as bioplastics packaging, trays, containers, disposal packaging, and food coating, as well as long-life applications such as plastic mulch, pharmaceutical uses, medicine, automotive, etc. 

Starch biopolymer is considered one of the most promising materials for replacing petroleum-based polymers, especially for food packaging, disposal packaging, and mulch film. This is due to its wide availability, renewability, biodegradability, and low-cost manufacturing compared to petrochemicals processes [5]. The use of biodegradable biopolymer starch has also been projected to minimize municipal waste and landfill operation costs, as starch biopolymers are fully biodegraded within two weeks [6,7,8]. Thermoplastic starch (TPS) can be obtained through the structural disruption occurring inside the starch granule when it is processed under the presence of shear, heat, and plasticizer. This would allow homogeneous melting of the thermoplastic starch to develop under these circumstances. Nevertheless, TPS has some drawbacks or weaknesses in terms of high water solubility, high moisture absorption and water barrier, and low flexibility. This is because of the abundance of hydroxyl groups in its structure. Hence, to overcome these limitation issues, much research has been conducted by scientists and engineers to enhance its properties, including graft copolymerization [9], chemical modification [10], incorporating fillers such as multi-walled carbon nanotubes [11], fibers [12,13], nanocellulose [6,7,8,14,15,16,17], cellulose [18], clay [19], nanoclay [20] and lignin [21], and blending with other synthetic polymers [22,23]. Reinforcing TPS with natural fibers is one of the most promising methods, as it is cost-effective, totally biodegradable, and enhances the mechanical properties of the composite materials [24,25,26]. The most common types of starch utilized for fabricating biopolymer include corn, wheat, rice, potato, sago, and cassava [27].

The properties and performance of natural fibers depend on several factors, such as their chemical composition, their geographical location, species, altitudes, growing conditions and harvesting times, preparations, extraction, processing, storage procedures, and treatments of natural fibers [28,29]. Natural plant fibers are generally referred to as lignocellulosic fibers, since most of them are cellulose fibrils incorporated in the lignin and hemicellulose matrix. The chemical composition of the plants varies from one plant to another [30]. In tropical regions, e.g., Malaysia, the Philippines, Indonesia, Nigeria, Thailand, Brazil, Indonesia, and the Democratic Republic of the Congo, cassava, scientifically known as *Manihot esculenta* Cranz, is the fourth most extensively produced starch product in the world after maize, potatoes, and wheat [31]. Thailand is the world’s largest cassava producer and exporter after Nigeria, which takes 50–75% of cassava starch’s global market share [32]. Cassava consists of approximately 60–70% starch content (dry basis) in granules form [32], lignocellulose (cellulose, hemicellulose, and lignin), and other secondary components (Table 1).

## 2. Natural Fiber and Natural Fiber Reinforced Composites (NFCs)

Natural fibers can be classified into three categories that are plants, animals, or minerals [38,39]. The widest utilization of natural fibers for reinforcement purposes is natural plant fibers [27,40,41]. Plant fibers are classified according to natural form or fiber-derived sections of the plant [42,43]. Figure 1 shows the different classes of plant fibers, (I) wood and (II) non-wood fibers (i.e., grass/reed, leaf, fruit, seed, wood stalk, and bast). The utilization of non-wood fibers for reinforcement with polymers has gained tremendous attention among researchers. Using these non-wood fibers helps conserve the natural forests, and excessive deforestation is worth discussing with the rising environmental issues. Tropical countries such as Malaysia, Thailand, Indonesia, and the Philippines have vast and unused natural fiber potential as an alternative to synthetic fibers. Cassava, cocoa pod husk, kenaf, sugar palm fibers, sago, pineapple leaf, oil palm fruit bunches and trunks, and coconut trunk fibers are various types of natural fibers that should be utilized and commercialized. Natural fibers are much more sustainable than synthetic ones. In production, cultivation, and manufacturing, they do not harm our environment [44,45]. Moreover, they are not hazardous to the health of the people who work with them. Moreover, even after processing, they are much more environmentally friendly than man-made fibers. Since synthetic fabrics like polyester release microplastics into the water during the washing process, synthetics harm not only nature and wildlife, but also humans with regard to the water cycle. Table 2 shows the advantages and disadvantages of natural fibers.

The rapid growth of the industrial revolution and global warming are pushing factories to manufacture products derived from natural resources [46,47,48]. Hence, a variety of crops are being produced extensively by cultivation, and preferences depend on the needs and values of societies. The crops are used for their natural polymers, such as starch or its fibers. Some of the plants with extensive cultivation are kenaf, sugar palm, pineapple, roselle, palm oil, etc. [49,50,51,52]. These fibers are produced to be reinforced with polymers in order to strengthen the mechanical and water barrier properties of the polymer composites. Research in polymer science and technology is responsible for manufacturing natural fibers and other highly usable agro-wastes globally.

The fabrication of natural fiber-reinforced polymer composites was expected to produce lighter composites with low cost production [53]. Natural fiber composites provide many advantages, such as biodegradability, availability, low density, and recyclability [54,55]. Bast fibers such as roselle, ramie, mulberry, okra, nettle, milkweed, linden, kudzu, kenaf, jute, hemp, flax, and sisal are the most common natural plants used in industrial applications such as packaging, construction, military, aerospace, medical, and automotive applications [56]. Natural fibers have been used extensively in biopolymer applications. This is because the bioplastics derived from natural sources have several hindrances, lower mechanical properties, as well as water resistivity, compared to fossil-source plastics [57]. Hence, it is of paramount importance to increase research efforts in this area considering the usage of natural materials obtained locally [58]. 

Natural fiber composites (NFCs) have been recently highlighted in various industrial applications and they are quietly replacing the utilization of conventional materials based on several factors [59]. The implementation of new materials in the industrial sector is usually hindered by constraints and limitations such as cost, compatibility with the product design, machinability, inherent relationship within the materials, and their availability, recyclability, and final product performance. This makes compromises between these constraints, advantages, and disadvantages in selecting materials an intricate matter, where proper decisions have to be made using modern techniques like optimization methods, informative decisions, and expert systems utilizing pairwise comparisons [60,61]. In comparison with conventional composites, NFCs have greater specific strength, fatigue, stiffness, non-toxicity, lower life-cycle cost, adaptability to hazardous environments, recyclability, greater impact absorption capacities, and better resistance to corrosion [62]. Such advantages of NFCs result from the advantages of their constituents (fillers and polymers), particularly that natural fibers themselves are better than traditional glass fibers in terms of good thermal and acoustical insulation characteristics, low cost, energy retrieval, availability, degradability, CO_2_ sequestration enhancements, reduced dermal and breathing discomfort, as well as lower tool wear in machining operations [62,63,64,65]. Natural fibers are obtained by processing waste from agriculture, industry, or consumers [2,7,8,15,30,66,67,68,69,70]. Moreover, some materials engineers are trying to manufacture safer and environmentally friendly plastics. To overcome the problem of non-biodegradable plastics and the disposal of agricultural waste, it is essential to manufacture environmentally friendly materials to offset the use of durable plastics [16,71,72,73,74]. The production of eco-friendly products from natural sources has minimized the dependency on traditional plastics, thus leading to solutions for environmental pollution issues. Recently, the utilization of natural fibers and agricultural by-products to develop biodegradable plastics, such as such as maize, sugar palms, potato tubers, kenaf, and cassava, has become increasingly of interest [75].

The characteristics and efficiency of NFC products depend upon the properties and compatibility of their individual components. Besides that, they also rely on the interfacial reinforcement of the natural fibers and polymer. This reinforcement extends the possibilities of manufacturing various innovative materials with completely new qualities [76,77,78,79,80,81]. However, there is output uncertainty associated with variation in the properties of natural fibers [82,83]. This requires careful study for selecting the highest-performance manufacturing for such types of composites under controlled conditions to achieve more reliable and better designed data.

## 3. Cassava (*Manihot esculenta*)

### 3.1. History of Cassava

Cassava is a tuberous, woody, and perennial plant; *Manihot esculenta* is in the family of the Euphorbiaceae (spurge family), and cassava plant and its roots are also known as yuca, manioc, and mandioca [84,85]. The term cassava originated from the word Cazabi or Casavi, which means bread in the Arawak language (the tongues of the first Indigenous communities living in the Great Antilles) [86]. This plant is available throughout the year because of its easy harvest process and being a periodical plant. These advantages make cassava a very reliable food crop. Cassava crop is rich in riboflavin, thiamine, and carbohydrates; however, it contains no protein [87]. The regions where cassava plants are grown in the world are Africa with 72%, Asia with 18%, and South America and North America with 10% [88]. Cassava yields are most important due to their many uses in industries such as bioethanol, starch, alcohol, and biofuels, as well as being used in the animal feed industry [89,90,91]. The Portuguese began cultivating cassava along the coast of West Africa, then introduced cassava to central Africa, eastern Africa, Ceylon, Madagascar, Malaya, India, and Indonesia. The cassava was introduced to Asia by the Spanish after their occupation of the Philippines was completed, and then spread throughout Asia by the nineteenth century [92,93,94]. Nigeria, Thailand, and Indonesia are now the leading producers of cassava in the world, with the production of 54.83, 30.02, and 23.44 million metric tons (MMT), respectively, in 2017 [95]. Cassava yields from other countries are as shown in Figure 2.

### 3.2. Cassava Plant Parts

Cassava (*Manihot esculenta*) can be found abundantly in various nations across Asia, Africa, and Latin America. The United Nations Food and Agriculture Organization (FAO) reported that cassava is placed fourth among developing countries’ food crops, after maize, rice, and wheat. Cassava roots have high starch content, and cassava is a very productive crop from the aspect of food calories generated per unit land area per day of 250,000 cal/hectare/day, as compared to maize, wheat, and rice, which are 200,000, 110,000, and 156,000, respectively. It is likewise utilized as a feedstock for various industrial applications [97]. The cassava plant is divided into three parts, such as leaves, stems and tubers, as appears in Figure 3.

Research interests in plant fibers are gradually increasing due to the benefits they offer, such as low cost, light weight, renewability, biodegradability, being environmentally friendly, recyclability, easy separation, and carbon dioxide sequestration ability [101]. Annual cassava root production around the world is estimated to be more than 250 million tons [102]. The biochemical compositions of cassava-based residues are presented in Table 3.

### 3.3. Cassava as Multipurpose Plant

Many researchers have recorded a significant number of uses for cassava. Modified cassava that is specially formulated for individual applications continues to find new uses every day. Cassava is traditionally processed in various ways to reduce its toxicity, improve its palatability, and convert the perishable fresh roots into more stable products [119]. Cassava is a versatile plant that offers various benefits, especially in food applications—cassava tuber is processed into chips and pellets that are edible [119]. Also, the roots and leaves are both used for animal feed [120]. In addition to that, cassava is very rich with starch, one of the sources of biodegradable polymers [121]. Cassava waste is a potential feedstock for integrated biorefineries for bioproducts co-production, e.g., glucose syrup (GS), succinic acid (SA), and bioethanol, as well as in combined heat and power (CHP) [122]. Treatments of the obtained glucose syrup from hydrolyzed cassava starch and cellulose with different enzymes yields a high fructose syrup [110].

### 3.4. Bio-Products from Cassava

Owing to the decline of fossil fuel resources and uncertainty in the Organisation of Petroleum Exporting Countries (OPEC), researchers are focusing on exploring sustainable and renewable energy sources [123,124,125]. The waste generated by the agro-industries constitutes a large share in the bioenergy production industry [126], including its consumption in ethanol and biofuel production, which is now a focal point of many researchers over the past decade, because of limited fossil oil reserves [127]. The production of biofuels from cassava starch began in the 1970s, and utilization of cassava residues such as peel, bagasse, stem, root, and leaf are currently being extensively studied by researchers to reveal their potential in biofuel production [95].

## 4. Cassava Starch

Starch is one of the most abundantly available and cheap agricultural products that is utterly biodegradable in many environments [3]. Starch is primarily derived from cereals such as wheat, maize, rice, as well as tubers such as cabbage and cassava. It is contained in seeds or roots and is the plant’s primary energy source [128]. Moreover, starch is also a polymer, which is a key factor for its various applications in many industries. The fabrication of thermoplastic starch includes extrusion and/or molding of temperature and pressure steps. Decomposition in the composting environment of the pure starch polymer occurred very quickly (a process that lasts about a month); however, it aged quite slowly and has no moisture resistance [129]. Production of cassava starch is very simple and facile by utilizing wet milling of fresh cassava. The following steps in Figure 4 show the stages of the production of cassava starch. Among the main factors when harvesting and choosing cassava starch’s root extract are the age and root quality. Cassava roots are highly perishable and the enzymatic cycle of rotting increases in a day or two; hence, immediate treatment after the harvesting is necessary. Treatments include peeling, rinsing, and converting cassava roots into fine particles. Next, a sifting step takes place to remove starch from the grated pulp, while the fibers remain intact. These fibers are then washed three to four times on a screen with distilled water. The extracted starch is left to sediment, and then all fibers are removed and the starch is rewashed with distilled water to get rid of any residue from the fibers. After that, the starch is oven fried for six hours at a constant temperature of 45 °C to reduce its moisture. The final drying process is performed under direct sunlight for four hours, and it is crushed using a dry blender prior to storage in an airtight container to prevent accumulation of moisture and contamination [119,130].

### Key Features of Cassava Starch

Cassava is an annual crop that is widely grown for its ability to produce both starch and alcohol. According to chemical analysis, cassava starch itself comprises 92.5 ± 0.9% (dry basis) carbohydrates, of which 83.5 ± 2.5% are starches. The remaining percentages are as follows: 10.2 ± 0.1% for moisture, 3.45 ± 0.04% for ashes, 0.48 ± 0.02% for lipids, 3.7 ± 0.7% for proteins, and 1.9 ± 0.2% for acid detergent fiber. Cassava starch thus has higher carbohydrate content compared to other lignocellulosic-derived natural starch. Regarding the amount of minerals present, cassava starch has dry basis contents of 43.8 ± 0.3% sodium, 49.8 ± 0.4% potassium, 61.6 ± 0.7% calcium, 43.4 ± 0.2% magnesium, and 26.0 ± 0.4% iron. Starch granules and irregular, distorted, truncated particles with low sphericity, which might primarily be connected to protein aggregates and fiber components, are morphological characteristics of cassava starch.

## 5. Cassava Fiber

### 5.1. Macro-Size Cassava Fiber

From the compositional analysis of cassava by a wt% dry basis, the crude fiber was 3.5 from unpeeled cassava (starch and peels), 2.0 from cassava starch, and 10.6 from cassava peel [117]. Also, the fiber of cassava that remained from another study was 23.30 wt%, dry basis [108]. The chemical compositions of cassava tubers are shown in Table 4. 

The fibrous residue from cassava bagasse contains around 50% starch on a dry weight basis [132], and its compositions are shown in Table 5. These analyses were performed on bagasse samples collected from different processing units and times in Parana state, Brazil.

### 5.2. Nano-Size Cassava Fiber (Preparation and Characterization of Nanocellulose from Cassava)

The broad plastics applications around the world and their inability to degrade resulted in white pollution that affected the world’s ecosystem [137]. Biodegradable agro-industry waste, including bagasse of sugar cane, malt, and cassava, as well as starches, can be used in the production or reinforcement of films in packaging products manufacturing [6,138]. Cassava bagasse can be employed in various higher-value applications, e.g., biodegradable packaging, as well as organic acid, nanoparticles, ethanol, biofuel, nanofibers, and a-amylase lactic acid productions, etc. [139]. Cassava bagasse also contains residual starch, fibers, 37% hemicellulose, and 38% cellulose and lignin [140]. Cassava peel and bagasse are being produced in huge quantities by the cassava starch industry. The peel that contributes 15% of the weight from cassava root is very rarely used and is wasted, whereas the bagasse is usually used in animal feed applications. The cellulose available in cassava peel is approximately 40–55% of the dried peel [141,142]; thus, it can be a source of cellulose.

Cassava peel (CP) was used to extract and prepare the cellulose and cellulose nanofiber (CNF), as conducted by Travalini et al. [143], via alkali treatment and bleaching process, as well as by mechanical disruption processes: homogenization and ultrasonication, for cellulose and cellulose nanofiber, respectively. The results collected were contrasted with the normal acid hydrolysis approach, and it was found that the CNF had average diameters of 8.2 nm and 6.7 nm, respectively, after homogenization and ultrasonication, as shown in Figure 5. In order to analyze the resulting cellulose and CNF from both processes, Fourier Transform Infrared (FTIR), X-Ray Diffraction (XRD), Scanning Electron Microscopy (SEM), Transmission Electron Microscopy (TEM), and Thermogravimetric Analysis (TGA) were used. The results showed that both procedures were successful in preparation of cellulose and CNF. Despite having similar chemical properties, CNF exhibited different physical properties than others.

## 6. Development of Biopolymer from Cassava Starch

### 6.1. Natural Fiber-Reinforced Cassava Starch Biopolymer

A study conducted by Ramirez et al. [144] on the reinforcement of green coconut fiber with cassava starch showed that the tensile properties were increased, and Young’s modulus was decreased with the decreasing of the diameter of the fiber, while the percentage elongation remained constant. The biocomposites were prepared using a compression molding technique with fiber concentration of 0, 5, 10, and 15%. The lignin content was found to be 35%. In addition, the moisture tension of biocomposites influenced the strength and elongation percentage. From a recent study, it was found that the cassava starch composites exhibited higher water absorption. 

Cassava starch has received growing attention as a biobased polymer, which is due to its renewability, cheap costs, availability, and being fully biodegradable in nature. Furthermore, cassava starch is an extremely viable candidate for sustainable materials production. Table 6 displays recent work on various natural fiber-reinforced cassava starch-based composites. For instance, Kaisangsri et al. [145] detailed the development of foam trays derived from cassava starch blended with natural polymers of fiber plus chitosan. Prior to incorporating chitosan solution into the batter at 0, 2, 4, and 6% (*w*/*v*) at 1:1 ratio, the kraft fiber was blended with 0, 10, 20, 30, and 40% (*w*/*w* of starch) of cassava starch solution. The hot mold baking process was then applied to prepare the cassava starch-based foam for 5 min in an oven at a controlled temperature of 250 °C. The findings have shown that the cassava starch foam formed with 30% kraft fiber and 4% chitosan exhibited identical properties to polystyrene foam. There was a small rise in color like L*, a* and b* value of the foam tray. The starch-based foam fabricated displayed improved density, tensile strength, and elongation of 0.14 g/cm^3^, 944.40 kPa, and 2.43%, respectively. In addition, other properties such as water solubility index (WSI) and water absorption index (WAI) were observed to be higher than polystyrene foam.

Lomelí-Ramírez et al. [146] reported that any new materials development requires complete characterization data to discover their potential applications. In that direction, the cassava starch biocomposites preparation was reported as having earlier incorporated up to 30 wt.% of green coconut fibers from Brazil using thermal molding. The characteristics of the treated and untreated matrices with their composites, both physical and tensile, were also identified. XRD, FTIR, and thermal stability analysis were used to carry out structural studies. FTIR study showed that any fiber or glycerol levels in the matrix were not chemically influenced, nor had they altered the starch. The composites’ increased crystallinity with fiber content was observed in XRD study, while thermal study in TGA/DTA exhibited improved thermal stability when the amount of fiber incorporation was raised. In contrast, increased storage modulus, lower damping with increasing fiber content, and higher glass transition temperature were observed in DMTA study. The increased interfacial linkage between the matrix and the fiber could lead to these effects. Also, Vallejos et al. [147] evaluated the potential of the fibrous material obtained via ethanol-water fractionation of bagasse as thermoplastic starches reinforcement to boost their mechanical characteristics. The compounds were processed using matrices of corn and cassava starches, plasticized with 30 wt.% glycerin and mixtures of 0, 5, 10, and 15 wt.% bagasse fiber at 150 °C in a rheometer. This step was following ASTM D638 method where the combinations were pressed on a hot plate press at 155 °C. The images of the composites were taken from XRD and SEM. Moisture absorption test had been done for 24 h at a temperature range of 20 °C to 23 °C with humidity of 53% RH, followed the standard of ASTM E104. Moreover, tensile tests and dynamic-mechanical analyses (DMA) were also carried out. Increased tensile strength by 44% and 47% in comparison with corn and cassava starches, respectively, were observed in fibers reinforced with 10 wt.% bagasse fiber. The reinforcement of 15 wt.% bagasse fiber raised the elastic modulus of starch matrices more than fourfold. The storage modulus obtained at 30 °C (*E*30 °C′) increased with the bagasse fiber content, following the tensile elastic modulus trend. The results proved that these fibers are potential raw materials for biodegradable composite materials’ development.

The potential of cellulose nanocrystals (CNCs) extracted from kenaf fibers was highlighted by Zainuddin et al. [148] for a reinforcing fillers application in starch-based biocomposites. CNCs with diameter of 12 nm were collected from hydrolyzation method of kenaf fibers with 65 wt.% sulfuric acids. On the other hand, cassava starch biocomposites were prepared via solution casting method with 0 wt.% to 10 wt.% of kenaf CNC fillers and glycerol/sorbitol (ratio of 50:50) as plasticizer. In order to characterize the sample, various techniques had been proposed and carried out, including tensile tests for mechanical properties, various microscopy analysis for morphology, as well as physical properties. From the obtained SEM images, CNCs appeared in white with shiny dots which was attributed to the good dispersion of nanofillers within the starchy matrix. The biocomposite films had relatively better mechanical strength than the unfilled starch films. Moreover, both pure matrix and 6 wt.% CNC biocomposites exhibited thin, transparent, and flexible properties, proving that the transparency of the film was not influenced by the CNCs loading. Furthermore, CNCs loading led to a decreasing water sensitivity.

A fibrous residue rich in non-extracted starch (bagasse) was found in the industrial manufacturing of cassava starch-based composites, similar to cardboard [149]. The production method used was identical to the small-scale artisan recycled papers; 90% cassava bagasse mixed with 10% of kraft paper (as long fibers source) were employed to manufacture these composites with improved mechanical properties. The process yielded composites with similar characteristics to the recycled paper molded fiber packaging used in egg boxes. Cassava bagasse, however, has more benefits over recycled paper because it originated from sustainable and known sources, as shown via water direct contact test by full immersion of both impregnated and non-impregnated materials in water and tensile strength. Slight water resistance was observed in the cassava bagasse-kraft paper composites. Meanwhile, the mass of water absorbed by the starch acetate-impregnated materials was approximately half that of the non-impregnated materials. However, in terms of tensile strength, the impregnation did not significantly influence the materials’ strength. Therefore, starch acetate is an excellent additive for waterproof materials applications, such as disposable trays.

Souza et al. [150] studied the influence of *Penicillium commune* and *Eurotium amstelodami* on the antimicrobial potential, barrier properties, and mechanical strength of cassava starch composite films incorporating cinnamon essential oil. From the study, they confirmed that the development of active packaging occurred with the incorporation of cinnamon in cassava starch films compared to the control film without antimicrobial agents. ANOVA (*p* > 0.05) test results indicated that the cinnamon essential oil loading affected the properties of the films with effective antimicrobial protection against *P. commune* and *E. amstelodami*, both common used fungi in bread production.

Solution casting method was used by Pierre et al. [151] in the preparation of composite films using indigenous glycerol-plasticized cassava starch and 2D or 3D synthetic fillers, e.g., Na-beidellite type 2:1 phyllosilicate and Beta zeolite. The composites’ filler contents effect and types of mechanical and barrier properties were given special attention. The lyophilized Beta zeolite-reinforced films demonstrated higher water vapor permeability (WVP) and water solubility (WS) values compared to pure starch. In contrast, the Na-beidellite or the non-lyophilized Beta zeolite composites showed improvement in WVP. For both filler types, a drastic rise in the mechanical properties, particularly in Young’s modulus, was observed.

**Table 6 materials-15-06992-t006:** Reported works on natural fiber-reinforced cassava starch-based composites.

Polymer Matrix	Fiber	Reference
Cassava starch	Green coconut fibers	[144]
Cassava starch	Coconut fibers	[145]
Cassava starch	Cassava bagasse	[147]
Cassava starch	Cellulose cassava bagasse nanofibrils (CBN)	[152]
Cassava starch	Cassava bagasse-kraft paper	[149]
Cassava starch	Cellulose nanocrystals from kenaf fibers	[150]
Native cassava starch	Cinnamon essential oil/clove essential oil/sucrose ester of fatty acids/sugar	[151]
Cassava starch	Kapok fiber	[153]
Cassava starch	Jute fiber	[153]
Cassava starch	Kaolinite-rich clay	[154]
Cassava starch	The exploitation of chitosan as a compatible malt	[155]
Cassava starch	Malt bagasse	[156]
Cassava starch	Blended with zein, gluten, soy protein, kraft fiber, and palm	[157]
Cassava flour (CF)/wheat flour (WF)	Cassava stillage residue (CSR)	[158] [159]
Poly(vinyl chloride) (PVC)		[160]
Final egg albumen: Cassava starch: sunflower oil		[161]
Cassava stillage residue (CSR)	Self-reinforced	[141]
Cassava Starch	Cassava peel/cassava bagasse	[156]
Cassava Starch	The remaining fibrous residue of cassava starch extraction	[162]
Cassava starch	Cassava nanofiber	[143]
Cassava starch	Cassava/sugar palm fiber	[71]
Cassava starch	Cassava bagasse cellulose nanofibrils	[163]
Cassava starch	Microcrystalline Cellulose Avicel PH101	[164]
Cassava starch	Rice husk fiber	[165]
Cassava starch	Rice husk fiber nanocrystalline cellulose	[165]
Cassava starch	Cassava root	[166]
Cassava starch	Cassava bagasse	[166]
Cassava starch	Cassava bagasse lignocellulose nanofibers (LCNF)/nanoclay (Nclay)	[143]
Cassava starch	Cassava bagasse/kraft paper	[149]
Cassava starch	Waxy starch nanocrystal	[167]
Cassava starch	Cassava bagasse	[31]
Cassava starch	Zinc oxide nanofiller	[168]
Cassava starch	Acetobacter xylinum bacterial cellulose (BC)	[169]
Cassava starch	Recycled newspaper pulp fiber	[170]
Cassava starch	Kenaf cellulose nanocrystals (CNCs)	[148]
Cassava starch	Montmorillonite	[171]
Cassava starch	Bamboo nanofibers	[171]
Cassava starch	Bacterial cellulose	[172]
Cassava starch	ZnO/bacterial cellulose	[173]
Cassava starch	Carnauba wax/cashew tree gum-based films	[174]
Cassava starch	Concentrated natural rubber latex/cotton fiber	[175]
Cassava starch	Cellulose pulp fibers modified with deposition of silica (SiO_2_) nanoparticles	[176]
Cassava starch	Cellulose fiber	[177]
Cassava starch	Cassava bagasse (CB)	[178]
Cassava starch	Cassava peel (CP)	[178]
Cassava starch	Coconut nanocellulose	[179]
Cassava starch	Licuri nanocellulose	[179]
Cassava starch	Corn stover nanocellulose	[179]
Cassava starch	Pulp of eucalyptus commercial nanocellulose	[179]
Cassava starch	Cassava peel	[180]
Cassava starch	Kenaf cellulose nanocrystals	[181]
Cassava starch	Oregano essential oil/sugarcane bagasse	[182]
Cassava starch	Nanoclay	[183]
Cassava starch	Zein oil	[184]
Cassava starch	Gluten oil	[184]
Cassava starch	Soy protein oil	[184]
Cassava starch	Kraft fiber oil	[184]
Cassava starch	Palm oil	[184]
Cassava starch	Kenaf nanocrystalline cellulose	[148]
Cassava starch	Brazilian coconut fiber	[144]
Cassava starch	Eucalyptus cellulose nanocrystals	[185]
Cassava starch	Nanoclay	[186]
Cassava starch	Cassava roots bagasse	[187]
Cassava starch	Cellulose fiber/nanoclay	[188]
Cassava starch	Sugarcane bagasse fibers/montmorillonite	[189]
Cassava starch	Sisal cellulose nanofibers	[190]
Cassava starch	Banana fibers	[191]
Cassava starch	Pineapple shell fiber	[192]
Cassava starch	Soybean hulls fiber	[193]
Cassava starch	Soybean hulls microcrystalline cellulose	[193]
Cassava starch	Polylactic acid	[194]
Cassava starch	Ramie fibers CNF/nano PCC tapioca starch	[195]
Cassava starch	Zeolite	[151]
Cassava starch	Beidellite	[151]
Cassava starch	Starch nanocrystals (SNCs)	[196]
Cassava starch	Nanofiber straw/ZnO	[197]
Cassava starch	Pectin particles	[198]
Cassava starch	Cotton fibers	[198]
Cassava starch	Ramie cellulose microfibrils	[199]
Cassava starch	Coconut fiber nanocellulose	[200]
Cassava starch	Bamboo fiber, lime juice, epoxidized waste cooking oil	[201]
Cassava starch	Cassava nanofibril	[138]
Cassava starch	Carboxymethylcellulose/lactic acid bacteria	[202]
Cassava starch	ZnO nanorods/PVA electrospun mats/rosemary extract	[203]
Cassava starch	Polyaniline	[204]
Cassava starch	Nanosilica (SiO_2_)	[205]
Cassava starch	Montmorillonite	[206]
Cassava starch	Silica	[207]
Cassava starch	Sisal fiber	[208]
Cassava starch	Cassava bagasse	[209]
Cassava starch	Coconut residue fiber	[210]
Cassava starch	Cassava cellulose nanocrystals	[211]
Cassava starch	Grape stalks	[212]
Cassava starch	Carboxymethylcellulose/turmeric oil	[213]
Cassava starch	Sisal fiber/carnauba wax	[214]
Cassava starch	Coconut fibers	[215]
Cassava starch	Kaolinite	[216]

Recently, investigators such as Wahyuningtiyas and Suryanto [183] have studied the influence of reinforcement of nanoclay within cassava starch, where glycerol had been used as a plasticizer. The properties of cassava starch-based bioplastic such as biodegradability, moisture absorption, tensile test, and crystallinity test of XRD were identified. The results displayed that the incorporation of nanoclay into cassava bioplastic increased the tensile strength and elongation properties of the bioplastic from 5.2 MPa to 6.3 MPa, and 11.9 to 13.5%, respectively. The biocomposites also showed an improvement in water absorption. Besides that, this research also showed that nanoclay-reinforced bioplastic was completely degraded within the sixth day. Wahyuningtiyas and Suryanto [183] concluded that cassava starch reinforced with nanoclay demonstrated good potential to be utilized as a bioplastic from renewable resources.

Besides that, Pierre et al. [151] determined the effect of Beta zeolite and Na-beidellite as filler content on the tensile and barrier properties of the cassava starch composites film, which was fabricated via casting method with glycerol as the plasticizer. Further analysis showed that the water solubility and water vapor permeability of lyophilized Beta zeolite-reinforced cassava starch films were higher than the neat cassava starch. Interestingly, Na-beidellite and non-lyophilized Beta zeolite-reinforced composite film indicated an improvement in the water vapor permeability as well as their tensile strength. The next section of the results was concerned with the increase of Na-beidellite and non-lyophilized Beta zeolite contents, and elongation at break of the composites, while a decreasing trend was recorded in tensile strength from 2.3 to 2 MPa, and from 2.0 to 1.5 MPa, respectively, as well as their Young’s modulus, which was 35 to 20 MPa and 20 to 15 MPa, respectively. This can be seen in Figure 6a–c due to the incompatible polymeric composite, which also led to poor interfacial adhesion between the reinforced phase and the matrix [151].

For better environmental consideration and sustainability, Amni et al. [197] worked on the reinforcement of cassava starch with nanofiber straw/ZnO. Distilled water was used as the solvent to fabricate the bioplastic film. The loading effect of fillers load, nanofiber straw, ZnO, or both, on the mechanical strength, water absorption, and the decomposition rate of the bioplastic films also were examined. In summary, the results provided important insights, in which the highest tensile strength of 3.2 MPa was obtained at 9% ZnO, while the highest elongation recorded was 34% at 1% nanofiber straw. Other than that, the highest water absorption was 27.23%, obtained at 1% of nanofiber straw. Based on Amni et al. [197], the bioplastic films were buried in the ground for 20 to 30 days. Also, Arrieta-Almario et al. [204] carried out a number of investigations into polyaniline-reinforced cassava starch composite materials. Glycerol, glutaraldehyde, and polyethylene glycol were used as plasticizers in this work. Further analysis showed that a dark biopolymer film with good mechanical strength was fabricated. This was owing to the hydrogen bonding between the polymer structures established between the OH-group originated from cassava starch with the NH-group from polyaniline. In addition, the reinforcement of conducting polymer polyaniline to the starch had improved their electrochemical properties. Thus, this composite material is suitable to be applied for electrochemical sensors and accumulators.

Liu, Fan, Pang, et al. [205] studied the effect of tensile action towards the structure and properties of thermoplastic nanosilica (SiO_2_)-reinforced cassava starch. The properties of the composite were studied during the retrogradation stages. From DSC analysis, the retrogradation enthalpy of the composite with stress was higher than the composite without the tensile action during the retrodegradation phase. Other than that, the decomposition temperature and activation energy of thermal degradation of TPS/SiO_2_ composite with tensile action were much higher than the neat composite. Another important finding was that the stress-strain curves represented the mechanical properties of the reinforced cassava starch composite that was enhanced as the retrogradation time increased. The composite exhibited the Maltese cross-pattern as analyzed from the polarized light microscopy. According to Huang, Han, et al. [206], cassava starch reinforced with montmorillonite was manufactured and its barrier properties were investigated. The montmorillonite was modified with ultrasonic, magnetization conditions, and organic modifiers, as the intercalation reaction was used to improve its barrier performance. Thus, the findings showed that, owing to the incorporation of montmorillonite, the transmittance of the composite film decreased to 600 nm in the visible region and also greatly prevented UV-light transmission. As stated in the work, the temperature of the composite film decomposition ranged from 200 °C to 500 °C, while its weight loss rate was around 80%. Further analysis indicated that the oxygen resistance capacity of the composite film was nearly zero, while the oxygen permeability of organic montmorillonite-reinforced cassava starch film (0.067 cm^3^/m^2^d) was lower than the composite film without magnetization (0.097 cm^3^/m^2^d). In addition, Figure 7 demonstrates that the tensile strength of the non-magnetized composite film increased from 5 MPa to 6.5 MPa, which was contributed by the higher montmorillonite loading content. Hence, it was observed that the introduction of Fe to the magnetized composite films did not demonstrate any enhancement in its mechanical strength.

### 6.2. Thermoplastic Cassava Starch Biopolymer Blend-Reinforced Natural Fibers

Polymer blends containing varied amounts of starch blended with other types of polymers were extensively studied as possible replacements for plastics, mainly in the area of packaging, as listed in Table 7. Starch by itself is unsuitable because of various disadvantages, including (I) brittleness in the absence of suitable plasticizers, (II) poor water resistance due to the hydrophilic nature of starch, (III) weak and soft nature of starch biopolymer in the presence of plasticizer, and (IV) weakening mechanical properties upon exposure to high humidity environmental conditions. Therefore, starch biopolymer needs to be blended and reinforced with other polymers and fibers in order to overcome these disadvantages. Nevertheless, several studies have been reported about the thermoplastic starch biopolymer blend-reinforced natural fibers, especially with cassava starch biopolymer. For example, Chotiprayon et al. [217] studied the effect of reinforcement of coir fiber in cassava starch/poly(lactic acid) blend on the mechanical strength. The TPS/PLA/CF biocomposites exhibited good mechanical properties due to stronger hydrogen bonding occurring between the cassava starch with PLA and/or the coir fibers, as well as higher PLA crystallinity. From the recorded data, the tensile strength of TPS/PLA (31.3 MPa) was higher than the TPS/PLA/CF composite (with a range of 26.6 MPa to 30.7 MPa). While Young’s modulus and elongation at break for the polymer composites ranged from 344.0 MPa to 424.7 MPa, and 2.1% to 2.4%, respectively, these were much lower than those of the TPS/PLA blend with 303.9 MPa and 4.1%, respectively. However, the increasing coir fiber loading within the biocomposites contributed to decreased melt flow ability and shear-thinning effect, and increased shear viscosity.

Abral et al. [225] studied the effect of cellulose fibers content in polyvinyl alcohol (PVA)-reinforced cassava starch composites on their thermal and moisture resistance, and mechanical strength. The PVA/S composites were fabricated via ultrasonic probe treatment where the starch was mixed with the PVA gel and short bacterial cellulose fibers. In the report, it was stated that the sonicated biocomposites possessed low thermal and moisture resistance and low transparency. On the other hand, the sonicated blend film exhibited lower tensile strength (5.5 MPa) than that of the non-sonicated PVA/S film (10.6 MPa). However, the increased fiber loading had improved the tensile strength of the sonicated PVA/S. It was recorded that the mechanical strength of PVA/S-10U (about 17 MPa) was higher than the non-sonicated one, PVA/S-10N (15 MPa), both with 10 g fibers addition. This was contributed by the strong hydrogen bond between the fiber and the matrix. Next, Lisdayana et al. [228] reported the effect of oil palm empty fruit nanocellulose-reinforced modified cassava, corn, and sago starch/PVA biocomposite films. Prior to the reinforcement, the oil palm empty fruit bunches were ground mechanically using an ultrafine grinder in order to extract the nanocellulose. Next, the evaporation casting method was used to fabricate the biocomposite films. What can be clearly seen in this report was the continual increase of nanocellulose-reinforced TPS/PVA blend tensile strength related to the increasing of nanocellulose content (0, 1, and 3% nanocellulose addition). For cassava starch, the nanocomposite film exhibited increasing tensile strength from 0.5 MPa to 3 MPa; however, the biocomposite films of corn and sago starch had a higher tensile strength of 5 MPa and 4.5 MPa, respectively, at 3% nanocellulose loading. This was due to the difference of amylose content in the starch.

Silviana and Subagio [229] studied the characterization of cassava bagasse starch-reinforced bamboo cellulose microfiber (MFC). In this work, epoxidized waste cooking oil (EWCO) and glycerol were utilized as plasticizer while lime juice was used as the cross linker. The author determined the mechanical, thermal, structural, as well as the crystallinity properties of the biocomposite. The composite with the composition of MFC 1%-*w*/*w*, EWCO of 0.125%-*v*/*v*, glycerol of 0.25%-*v*/*v*, and lime juice of 0.125%-*v*/*v*, exhibited the highest tensile strength of 25.8 MPa. For thermal analysis, the weight loss and temperature of degradation of the modified cassava starch were approximately 22% and 290 °C, respectively. The reinforcement had enhanced the mechanical, thermal, as well as structural properties of the polymer. Likewise, Subramanya and Prabhakara [230] fabricated banana fiber-reinforced cassava starch biocomposites via alkali treatment and hot compression technique. The authors studied the effect of fiber lengths and fiber volume fractions on the tensile and impact strength of the biocomposite. The results displayed that the mechanical properties of the biocomposites were enhanced with the fiber length and glycerol content; the tensile strength, tensile modulus, and impact strength of the 20 mm banana fiber/cassava starch composites with 15% of glycerol were higher when compared to those of the neat matrix composites, rising from 2.82 MPa to 10.12 MPa, from 49 MPa to 356 MPa, and from 4.03 KJ/m^2^ to 8.31 KJ/m^2^, respectively. This was due to the composites’ fibrillation, which comprised the breaking down of fiber bundles into smaller bundles, hence increasing the effective surface area of the biocomposites for matrix adhesion.

Yi et al. [231] investigated the consequences of cassava starch loading on the surface morphology, crystallinity, changes in the functional group, and thermal properties of the PBAT/cassava starch/nano-ZnO composites films. The increasing of thermoplastic cassava starch loading improved the water absorption and lessened the water vapor resistance, mean water contact angle, as well as the light transmittance. Besides that, the addition of nano-ZnO decreased the water contact angle of the biocomposites; however, it did improve the composite film with 10% starch and 1% of ZnO at 95.5°. Other than that, the elongation at break and the tensile strength of the film were reduced as the starch content was increased. Moreover, the higher nano-ZnO loading enhanced the mechanical properties of the composites; the 10% TPS-reinforced PBAT composite films with the addition of 1% nano-ZnO had higher tensile strength (26 MPa) and elongation at break (530%) than the pristine composite films. 

The structure and surface of a composite film fabricated from extrusion of cassava starch with PLA and biodegraded by compost conditions were assessed by del Rosario Salazar-Sánchez et al. [233]. It was observed that the carbonyl index increased as the PLA content increased. Besides that, the higher carbonyl index in the composites could be contributed by the inclusion of anhydrous malic acid in PLA/TPS mixtures. The TPS/PLA biocomposite films were constructed through three main phases of biodegradation: breakdown, fragmentation, and the formation of minerals. Within week four, the film was observed to have biodegraded by 65%. 

### 6.3. Cassava Starch Hybrid Polymer Composites

Hybrid composite is referred to as the product which was combined into a single matrix by two or more different fibers, in order to achieve better mechanical, thermal, and barrier properties than individual fibers [50,51,52,234]. The properties of the hybrid composite are mostly dependent on the loading and orientation of the fibers, types of polymer matrix and fibers, and the degree of intermingling of the two or more fibers. Besides that, the mechanical properties of the hybrid composite are dependent on the failure strain of each fiber as well as the interaction between the polymer and fibers, in which the optimum hybrid performance is achieved when the mechanical strain of both fibers is highly compatible [235]. Nevertheless, numerous studies on the hybridization of composites, in particular cassava polymer and fibers, have been published, as displayed in Table 8. For instance, Edhirej, Sapuan, Jawaid, and Zahari [71] worked on sugar palm fiber (SPF)-reinforced cassava starch/cassava bagasse. The biocomposite films were fabricated via casting method and plasticized with fructose. In their work, the amount of SPF loadings was varied by 2, 4, 6, and 8% *w*/*w* of dry starch, mixed with the CS/CB biocomposites. In addition, the authors claimed that the reinforcement of SPF had relatively enhanced the physical and mechanical properties, as well as reducing the water uptake, water content and water solubility, and the density of the composite films. According to SEM images, the filler was completely incorporated within the starch matrices. Hence, the film with higher SPF content, 8% *w*/*w* SPF/CS/CB, exhibited better heterogeneous film surface. Moreover, XRD analysis proved hybrid film modified with SPF had improved crystallinity up to 47% compared to 32% for the pristine starch film.

Travalini et al. [143] studied the cassava bagasse-based nanofibers-reinforced cassava starch composite films. Initially, lignocellulose nanofibers (LCNF) were extracted from cassava bagasse via enzyme treatment prior to the colloidal mill, which were then utilized as a reinforcing agent in the cassava composite films. The authors focused on the effect of nanofibers loading (0.65 and 1.3% *w*/*w* loading) in starch-based biocomposite films on their mechanical, structural, as well as thermal stability, compared to the neat commercialized nanoclay. The results showed that cassava starch-based films were transparent and flexible. The better nanoparticles distribution within the film indicated that the nanofibers were well-incorporated within the cassava starch matrix, potentially applicable for packaging uses. On the other hand, the films’ opacity and water uptake values were dramatically decreased (0.65% for LCNF reinforced film and 1.3% for the nanoclay), whereas a lower concentration of LCNF resulted in the lowest WVP value. Next, the tensile stress of the nanocellulose reinforced starch-based films (6.6 MPa for 1.3% loading) was improved relative to the pristine CS (4.8 MPa). Likewise, Basuki et al. [173] worked on the effect of adding ZnO into bacterial cellulose reinforced with cassava starch biofoam on its crystallinity and water uptake capacity. The highly hydrophilic biofoam was synthesized via the baking process with various ZnO content (0, 3, 6, and 9%) in the bacterial cellulose-reinforced starch biofoam. Results showed that the ZnO addition had reduced the water absorption of the composite films up to 33% relative to the 0% ZnO content starch film, as shown in Figure 8. Moreover, the composite with 9% ZnO concentration exhibited the lowest water absorption amount of 0.164 (gH_2_O/g). XRD analysis indicated the improvement in the crystallinity, diffraction position, and intensity of the biofoam due to the increasing ZnO concentration. 

A study on the effect of the reinforcement of cellulose fibers and/or nanoclay on mechanical, structural, and water vapor resistance properties of the potato and cassava starches composite films was performed [188]. The biocomposite films were fabricated with starch, glycerol, cellulose fiber, and/or nanoclay, each with the content of 4%, 30%, 20%, and 5%, respectively. The physical, water vapor resistance, tensile strength, and morphological properties of the potato and starch-based films were measured. El Halal et al. [188] had discovered that fiber-reinforced potato starch exhibited more resistant films with enhanced solubility compared to cassava starch-based composite films. Table 9 below shows the tensile strength and elongation at break of cassava and potato starch films with different amounts of reinforcing agents. Nevertheless, the inclusion of nanoclay and cellulose in cassava starch increased the tensile strength from 4.7 MPa to 11.7 MPa and reduced the elongation from 4.62% to 1.62% relative to the neat starch film. Besides that, as cellulose and nanoclay were concurrently applied to the starch film, the water vapor permeability for the films were reduced.

In addition, Syafri et al. [195] fabricated cassava starch modified with cellulose nanofibers (CNF) ramie hybrid composites. Prior to the reinforcement process, the CNF-ramie was produced via chemical-ultrasonication method, whereas the CNF-ramie reinforced starch was fabricated using the casting solution with glycerol as a plasticizer. From the findings, the CS/4%-CNF/6%-PCC hybrid film exhibited the highest tensile strength with 12.84 MPa, as well as 30.76% crystallinity. Hence, it proved that the inclusion of CNF-ramie and PPC had enhanced the crystallinity, water vapor uptake, thermal stability, and mechanical properties of the hybrid nanocomposites. Li et al. [202] discovered the reinforcement of lactic acid bacteria to cassava starch/carboxymethylcellulose (CMC) biocomposite films to enhance the shelf life of bananas. Their research described how two species of lactic acid bacteria (LAB) (*Lactobacillus plantarum* and *Pedocococcus pentosaceus*) from pickled water, with extremely high exopolysaccharide (EPS) yield, were selected. The authors fabricated LAB, sodium CMC, and glycerol as plasticizers in cassava starch biocomposite films. A varied amount of LAB concentrations (0.5, 1, 1.5, and 2%) were introduced to the starch to evaluate the effect on antioxidant performance and water vapor permeability of the films. Table 10 displays the tensile strength and elongation at break of CS/CMC/LPL and CS/CMC/PPE composite films with various probiotics loading. The antioxidant function of the composite films was considerably increased after the inclusion of probiotics, and the CS/CMC/LPL-2% sample recorded the highest antioxidant activity of 48.12%. *L. plantarum* demonstrated a consistent CS/CMC (CS/CMC/LPL) matrix distribution, which was a thicker structure and effectively prevented water molecules’ penetration and provided ultraviolet protection. Thus, it inhibited the degradation of lipid oxidation in food packaging. The usage of composite film reinforced with 2% LAB in food packaging qualitatively increased the shelf life of the banana.

## 7. Cassava Fiber-Reinforced Polymer Composites

Natural fibers, or lignocellulosic fibers, are being used in polymers and composites increasingly as reinforcement materials. Compared to inorganic fillers such as glass fibers and other synthetic fibers, lignocellulosic fibers are more significant due to their advantages such as high specific strength and modulus, low cost and density, wide availability, low energy consumption during the manufacturing process, biodegradability, their abundance, and renewable nature. Agricultural waste or agro-industrial by-products can be considered important for obtaining natural fibers. Polymer composites with cassava lignocellulosic fiber as reinforcement filler have been examined. Table 11 displays the reported work on cassava fiber-reinforced polymer composites. There has been a lot of work on the preparation and characterization of cassava fibers reinforced with various types of polymer composites. For instance, the thermo-mechanical properties of egg albumen–cassava starch composite films containing sunflower-oil droplets that were influenced by moisture content were studied by Wongsasulak et al. [160]. The moisture content effect on the structural and thermo-mechanical properties of the composite films was analyzed using SEM, differential scanning calorimetry (DSC), and dynamic mechanical analysis (DMA). The cold gelation method was used to fabricate the composite films and they were dried in a moisture-controlled incubator (83.5%RH) for eight days at 25 °C. The composites were stored at varying relative humidity at a room temperature of 21 °C for over seven days to yield composite films with a moisture content of 4%, 7%, 11%, 17%, and 46% (dry weight basis). In DMA thermograms, the magnitude of G” and G’ were raised with temperature in high-moisture samples, decreased, and were gradually increased again for medium-moisture samples, and decreased in low-moisture samples, respectively. Two distinct peaks (at 49–53 °C and 79.8 to 132.4 °C) were observed in DSC thermograms that were attributed to protein denaturation and phase changes. The changes in the microstructure of the composite matrix were exhibited in SEM images as a result of different heating temperature and moisture content. This study proved that moisture content is crucial in the determination of microstructure and thermo-mechanical properties of egg albumen–cassava starch composite films incorporated with sunflower oil.

Previous research by Lomelí-Ramírez et al. [146] established biocomposite films of green coconut fiber-reinforced cassava starch via thermal molding technique. Observations from the composites’ characterization on mechanical strengths like structural and tensile strength were collected via FTIR, TGA, and crystallinity test, and DMTA studies, respectively. The results displayed the effect of increased coir fiber addition, from 0 wt.% to 30 wt.%, to the starch. For instance, the FTIR analysis revealed that there was slight decomposition of components in the TPS matrix during the thermal molding process, as the presence of carbonyl compounds had been detected by a distinct signal at 1715 cm**^−^**^1^. Nevertheless, from the crystallinity test of XRD, the increasing amount of fiber content within the TPS matrix contributed to relatively improved composite crystallinity values, from 39% to 62%. Improvement in thermal stability had been exhibited by the composite films as the degradation temperature was reduced by the higher coir fiber contents, which led to increasing fiber incorporation within the film. Moreover, the thermo-mechanical studies had shown the improvement of tensile strength and storage modulus, from 2027 MPa to 3215 MPa, as well as higher glass transition temperature. Thus, it proved lower damping within the composite structure. 

As noted by Vallejos et al. [147], bagasse fiber-reinforced thermoplastic cassava starch exhibited higher tensile strength as well as thermal stability. The author had worked on the effect of bagasse fiber reinforcement with cassava and corn starch in order to enhance the biocomposites’ mechanical strength. The fiber was extracted from bagasse via ethanol-water fractionation; 30 wt.% of glycerin was used as a plasticizer in the study. The varied amounts of bagasse fiber, 0, 5, 10, and 15 wt.%, were introduced to the TPS, which were then mixed in a rheometer at constant temperature of 150 °C. Table 12 displays the dynamic-mechanical properties of cassava and corn thermoplastic starch and their composites. Under the temperature of 155 °C, the mixtures had been pressed using a hot plate press method. Mechanical, structural, crystallinity, and moisture absorption properties of the biocomposites were tested. The results have shown that the higher fiber loading enhanced the tensile strength of the TPS. However, the addition of more than 10 wt.% bagasse fiber did not affect its mechanical strength, which owed to the agglomeration of fibers in the matrix. Furthermore, increasing tensile strength had been indicated for 10 wt.% fiber loading, being 44% and 47% higher than that of the neat corn and cassava starches, respectively, while the addition of 15 wt.% bagasse fiber had enhanced more than fourfold the starch’s elastic modulus.

Wongsasulak et al. [160] studied the characterization of egg-albumen-cassava starch biocomposites with sunflower oil droplets. The effect of moisture content on the structural and thermo-mechanical properties of the composites was analyzed using SEM, differential scanning calorimetry (DSC), and dynamic mechanical analysis (DMA). Cold gelatin was utilized for the biocomposites’ preparation, and they were left in an incubator with constant 83.5% relative humidity and temperature of 25 °C for a whole week. Next, the biocomposites were stored at a different relative humidity (with moisture contents of 4, 7, 11, 17, and 46%) at ambient temperature. The results from SEM structural characterizations showed that the composite matrix microstructure was modified with different heating temperatures and also moisture contents. In addition, DSC thermograms revealed two distinguishable peaks at temperature ranges from 49 °C to 53 °C and from 80 °C to 132 °C, which were contributed by the phase transitions and protein denaturation, respectively. Thus, from the achieved results, the author claimed that moisture content affected the thermo-mechanical properties and microstructure characteristics of the sunflower oil modified egg-albumen-cassava starch biocomposite films. In addition, Nguyen et al. [237] explained the effect of the addition of compatibilizer, glycidyl methacrylate (GMA)-modified PLA (PLA-g-GMA), to PLA/CNF biocomposites. Initially, the CNFs were harvested from cassava pulp and mixed with PLA via the melt mixing method. Through the mechanical test, the results displayed the enhancement in impact strength, tensile modulus, and elongation at break for the PLA/PLA-g-GMA/CNF composite films with an increasing amount of fiber loading. Here, PLA-g-GMA acted as the plasticizer of the composite. The distinct values of the mechanical properties were obtained from biocomposites with 0.1 wt.% CNF and PLA-g-GMA. The 0.1 wt.% CNF/PLA-g-GMA/PLA exhibited higher tensile modulus, impact strength, and elongation at break values than the pristine PLA, with 59 MPa, 22 kJ/m^3^, and 10%, respectively, compared to 56 MPa, 16 kJ/m^3^, and 8%, respectively. Hence, the mechanical strength improvement of the composites was due to their higher crystallization contributed by the plasticizer.

In another work by Huang et al. [138], the modified CNF played the reinforcer and compatibilizer roles for thermoplastic cassava starch. Prior to mixing, malic acid and KH-550 silane coupling agent were introduced to the CNF, which then altered the hydroxyl groups. The modified CNF addition was used in order to enhance the mechanical, water barrier, and hydrophobic properties of the CNF/TPS biocomposite films. The author claimed that the inclusion of malic acid and silane KH-550 had led to higher thermal stability and dispersibility of the CNFs. Hence, the reinforcement of modified CNFs enhanced the mechanical strength, hydrophobicity, and water vapor permeability (WVP) of the modified CNF/TPS composite films (25 MPa, 6% of water absorption, and 2.5 g.cm/(cm^2^.s.Pa) WVP) compared to the neat CNF/TPS (7 MPa, 7% of water absorption, and 2.9 g.cm/(cm^2^.s.Pa) of WVP), by 1034%, 129.4%, and 35.95%, respectively. Furthermore, Versino and García [187] determined the effect of cassava bagasse particle size on the mechanical and structural properties of cassava bagasse-reinforced cassava starch composite film. The films were fabricated via casting molding the gelatinized starch suspensions with 1.5% *w*/*w* bagasse, while glycerol was used as the composite’s plasticizer. Different fibrous residue fractions, with particle sizes ranging from 500 to 250 μm, 250 to 53 μm, and below 53 μm, were introduced and compared to 500 μm-sized bagasse particles-reinforced composite film. In addition, cassava bagasse’s chemical composition and distribution of particle size aided in explaining the changes in morphology, barrier properties, and mechanical strength of the starch-based films. SEM images displayed that the filler was structurally integrated into the matrix. Thus, the tensile mechanical resistance and elastic modulus of the bagasse-reinforced cassava starch, 14.7 MPa and 1247 MPa, respectively, were higher compared to the neat film with 1.3 MPa and 505 MPa, respectively. The fabricated biocomposite films are suitable for flexible packaging applications with a slow degradation process.

## 8. Potential Applications

Natural fiber-reinforced polymer composites, or biocomposites, have been a center of interest for the research community for the past few decades [27]. These biocomposites are proposed to replace synthetic-based fibers because of their excellent attributes, including low cost, light weight, biodegradability, high specific properties, less abrasiveness to equipment, being environmentally-friendly, and their sustainability as well as renewability, hence producing a positive environmental impact [238,239,240,241,242,243]. Amongst the natural fibers produced, cassava is one of the most important natural fibers, and has been generally recognized as a possible reinforcement in polymer composites by the composite industry. Compared with synthetic fibers, cassava fiber has lower thermal resistance, similar to other natural fibers.

Therefore, according to the benefit of cassava fiber-reinforced polymer composites, as mentioned, cassava fiber-reinforced polymer composites have huge potential to be used in a wide range of various applications. Figure 9 displays the potential options for cassava fiber-reinforced polymer composite uses in various sectors, including construction and housing, safety, household items, textiles, pulp and paper, decoration, electronics, packaging, wood panels, as well as automotive applications. 

### 8.1. Automotive

Raw materials such as cellulose fibers (cassava, abaca, jute, hemp, sisal, kenaf, cotton), softwood, or hardwood fibers, are widely used in the automotive industry. Natural fibers offer a fascinating range of applications due to their promising properties. A very large amount of research has been performed to improve their mechanical, physical, thermal, and water barrier properties for boosting their applications [27,244]. Some of their drawbacks, including low mechanical strength due to incompatibility between the fiber with the matrix, as well as the hydrophilic nature of natural fibers, can be improved via specific pre-treatment, specifically chemical treatment such as alkali treatment, bleaching, acetylation, benzoylation, and others, which have been comprehensively discussed in other studies [30,50,51,52,245]. The new products made from composite materials are usually manufactured via extrusion or injection molding techniques. The automotive industry is interested in new materials to fulfill the new regulations that cars should be partially decomposable or recyclable [246].

Advanced technology using natural fiber-reinforced polymer composites in interior parts of automobiles is being manufactured. Most fabrication works are focused on polypropylene (PP)-based composites manufactured by commingled mats of PP and natural fibers, by thermoforming extruded sheets, or by compression molding. Natural fibers, except at extremely low temperatures, do not provide as much impact resistance as glass fiber. This is due to the properties of natural fibers themselves, which are confined to operating temperatures above 180 **°**C. Besides that, some natural fibers may emit unpleasant odors if care is not taken when the natural fibers are handled.

Natural fibers have lower specific gravities of 1.25–1.50 g/cm^3^ compared to glass fiber, which is 2.6 g/cm^3^. That helps give natural fibers a higher strength-to-weight ratio for reinforcing plastic polymers. In polyurethane composites, natural fibers are often used. The first commercial example is the inner-door panel for the 1999 Mercedes-Benz S-Class, assembled in Germany with a 35% semi-rigid Baypreg F PUR elastomer and a 65% flax, hemp, and sisal combination. The 2-mm door panel was created by Nafpur Tec process from Bayer’s Hennecke Machinery Unit, in which a robot put the natural fiber-based mat into an opened mold and a second robot poured PUR on it prior to closing. The procedure also was utilized to manufacture a European 2000-concept car sunroof cover. 

Currently, major car manufacturers are producing “vegan cars” with a wide range of creative, premium alternatives to leather [247]. Vegan leather is usually made from polyurethane. It can also be fabricated from renewable and sustainable materials such as cassava, apple peels, pineapple leaves, or other agro-waste, as well as recycled plastic. According to Hanson [247], the Model 3 Tesla is completely vegan. In addition, Audi also includes this material in their manufacturing, even though other luxury car manufacturers like Mercedes and BMW offer exclusive products: Artico and Sensatec, respectively. All manufacturers are increasingly trying to attract eco-conscious drivers, such as with Lexus’ NuLuxe, Land Rover’s SueDecloth, and Softex from Toyota. Most of them are mindful of the interconnected attraction of vegan and eco-conscious drivers, and they prefer to include these materials in their new electric designs. With a carbon-fiber frame and natural fibers in the interior, the BMW i3 is the only existing such electric vehicle by BMW.

### 8.2. Packaging

Plastic bags, commonly made of petroleum-based materials, are lightweight, versatile, reliable, and inexpensive, and they have become an important asset of our everyday life. This robust polymer has been used in every facet of our lives today; it is used to manufacture goods used every day including food packets, handsets, speakers, debit cards, vehicles, and even toothbrushes. These synthetic chemicals not only harm human health but also wreck the environment. Annually, about 8.3 billion metric tons of plastics are used for packaging-related applications such as shopping bags, electronic packaging, food packaging, waste bags, agricultural mulch films, etc. Of the annually produced plastics, those which end up becoming plastic waste amount to 6.3 billion metric tons. Merely 9% of it is recycled. The largest portion of them, 79%, accumulate in soils in the existing litter environment. Plus, most of the plastic waste can be found in the sea, which is the final sink. If patterns continue, 12 billion metric tons of plastic will be stored in the landfills by 2050. This would be 35,000 times more than the Empire State Building [248].

In order to mitigate the global crisis of non-biodegradable and non-compostable plastic bags, cassava biopolymer has been introduced. These bags are made from starch harvested from cassava plants, which are biodegradable and environmentally friendly. Its nature plays an important role in biodiversity conservation, saving aquatic species and the avoidance of ecological disasters. The plastic bags made from cassava do not harm the environment. These environmentally friendly bags consume less time to compost compared to petroleum-based plastic bags. In addition, the cassava bags are water soluble and do not contaminate the environment under high temperatures.

Currently, much research on utilizing cassava starch and fibers for packaging applications is being performed. For environmental waste management, the utilization of cassava starch films as a potential alternative packaging is essential in order to replace petroleum-based plastics. Recently, authors [31,71,178,180,249] have studied cassava-based films that were fabricated via solution casting methods, for food packaging.

The effects on physical, thermal, mechanical, and structural properties of cassava starch-based films was examined for various plasticizer types including fructose, urea, tri-ethylene glycol, and triethanolamine, and for different concentrations of dry starch, 0.30, 0.45, and 0.60 g/g [249]. Regardless of the plasticizer type, the moisture content, water solubility, and water absorption of the films improved with the plasticizer loading. Other than that, regardless of the plasticizer type, the glass transition temperatures of the films also reduced with higher loading of plasticizer. Reduced tensile strength yet greater elongation at break of the film samples was recorded, which is attributed to the increasing plasticizer concentration. The mechanical analysis displayed that the highest tensile strength (4.7 MPa) and Young’s modulus (69 MPa) had been recorded from the film plasticized with 30% fructose. The findings revealed that 30% fructose-reinforced SPS films showed improved mechanical properties and could improve food packaging. Furthermore, many attempts are being made to develop the functional properties of cassava-based films as an important material for food packaging.

There are three biodegradable cassava bag manufacturing steps: (1) granules production, where the starch extracted from cassava flour is blended with such plasticizing agents as glycerol and sorbitol. Next, an antimicrobial agent is applied to the starch, such as essential oils or silver nanoparticles, to counteract the assault from bacteria [250,251]. This starch blend is inserted in a granulator chamber, which results in the processing of granules at a given temperature and pressure. (2) Film production, where granules obtained in the previous step are placed into an extruder machine and the thickness of the films is adjusted based on the manufacturer’s requirements. This starch-based film is expanded by rollers. The film will then be refreshed and cooled, and rolled for use in the manufacture of cassava bags. The final step is (3) bag processing, which includes the preparation, with the aid of the film generated in the second step, of cassava-derived biopolymer bags. Various kinds of cassava bags, such as the T-shirt bag (Figure 10a), grip-hole bag (Figure 10b), and garbage bag (Figure 10c), are produced in this final step.

### 8.3. Food Coating

The edible coating is a fine film of edible substance that can be consumed as part of the food product. Edible coatings are good shields against harmful biological, physical, and chemical modifications to protect the product. They help it avoid moisture depletion and selectively permit respiratory gases such as CO_2_ or O_2_. Because of its strong mechanical properties, starch is one of the edible materials used as a film cover; it is isotropic, odorless, tasteless, colorless, and flexible. Bee wax was the very first comestible layer in the 12th century for oranges and lemons. Fats have been used in England to extend the shelf life of meat products, known as larding. These coatings were already used in the 20th century to avoid water loss and add a gloss layer to fruits and vegetables. Some of the essential properties of edible coatings are (I) they do not ferment, coagulate, differentiate, develop an off-flavor, or spoil during storage; (II) they spread evenly, dry fast, and are easy to remove from the machinery; (III) they should not crack, corrode, or peel during storage and processing; (IV) they should not be kept in a package and react to food; (V) they allow adequate gas exchange to avoid any occurrence of the flavors or spoil during storage; and (VI) the coating ought to be a moisture barrier in order to avoid soggyness [252,253]. There are numerous advantages of using edible coatings, such as they improve appearance and structural properties, as well as reducing water loss, gas diffusion, and mold growth [254]. Cassava starch suspension was prepared with constant stirring at 60 °C prior to a cooling process at ambient temperature. After that, the food to be coated was dipped in the solution and allowed to dry at room temperature [255]. The concept behind this technology is to create an altered atmosphere around the fruit surface that could maintain features of fruit quality. The ability of water-soluble polysaccharides in coated fruit and vegetables to decrease O_2_ and increase CO_2_ in the internal environment thus decreases the respiration rates. In a similar way, with controlled and customized atmosphere storage, the shelf-life of fresh items can be improved. There are several coating edible material methods such as dipping, dripping, fluidized bed coating, panning, and spraying. Gracia et al. [256] conducted a study on the effect of cassava starch coatings, with or without potassium sorbate, on the quality of processed strawberries. The findings revealed, with or without potassium sorbate, that the water vapor resistance test of cassava starch edible coatings had not affected the superficial color of the strawberries and demonstrated strong acceptability. In addition, the coatings also contributed to the decreased of rate of respiration and improved the susceptibility of samples to water vapor. Furthermore, Castricini et al. conducted investigations on the impact of cassava starch and carboxymethyl starch coating on the sensory properties of papaya when processed [256]. The result shows the edible coating of cassava rapidly increasing its shelf life by reducing the rate of respiration. It also decreases the loss of weight of food by eliminating juice leakage, and increases the quality of food by applying gloss to the food [257,258,259,260,261,262]. 

### 8.4. Renewable Energy (Bioethanol)

The rise in energy use and climate change have led to huge strides in the world‘s search for renewable energy sources. Moreover, bioethanol is a promising renewable fuel and is already mixed with gasoline in many countries. In 2017, it was estimated that bio-ethanol global production was 27.05 billion gallons, with the United States having contributed about 58%, which was 15.8 billion gallons [263]. The world’s economy is significantly dependent on fossil energy resources, where combustion contributes to approximately 98% of carbon emissions [264,265]. Bioethanol fuel is produced mainly through the fermentation of biomass sugar components, including cassava starch, cane juice, and other carbon sources [266,267]. One of the advantages of cassava is that it can be converted into value-added components, such as methane (biogas) and ethanol, which in most countries will be more than enough for people’s needs [268]. Next, cassava has the ability to grow on degraded lands and marginal soils and has high drought-tolerance, and it has the third-highest carbohydrate yield per one hectare after sugar beet and sugarcane [269].

Cassava has been widely used to produce bioethanol using cassava peel waste as feedstock for microbiological and enzymatic hydrolysis. This waste is rich in cellulose, which is about 43.626% [270]. Cassava starch-based bioethanol produced from fermentation, enzymatic hydrolysis, and ex situ nanofiltration have been studied. Cassava starch was liquefied and converted into sugars by gluco-amylase and alpha-amylase, respectively, prior to fermentation for bioethanol production. Response surface methodology (RSM) methodology was used to optimize the fluidization and sugar conditions on sugar concentrations [266]. Bioethanol is also produced from hybrid cassava peel and pulp via acid and microbial hydrolysis. The results were obtained from converting cassava peels to easily detect a more useful way to manage cassava waste in the environment [264]. The results were investigated and showed that cassava peels’ starch could be readily degraded, and bioethanol can be produced by Aspergillus Niger (AN). The bioethanol produced was similar to ethanol [271]. It was also done to produce bioethanol using cassava dough liquid waste as the enzyme stimulant in Gari processing. In West Africa, Gari is a delicate, nourishing powder made from the root of the cassava tuber, which in the manufacturing process is converted into edible dry granules [272]. An integrated system of continuous bioethanol production resulted in fuel-grade bioethanol with purity up to 99.8% (*w*/*w*), a technically and economically feasible venture to invest in [273]. Another study revealed that cassava and sweet potato peels were used as feedstock for bioethanol production with maximum yield via *Pleurotus ostreatus* and *Gloeophyllum sepiarium* hydrolysis and *Saccharomyces cerevisiae* and *Zymomonas mobilis* fermentation [274]. Highly concentrated bioethanol was produced by repeated hydrolysis and intermittent yeast inoculation from cassava stem. High-concentration bioethanol production (about 40 g/L) approaches were explored by investigating the effects of intermittent inoculation yeast on ethanol fermentation, hydrolysate concentration, and acid hydrolysis conditions [275]. Many studies have shown that bioethanol can be extracted from the cassava plant parts, which are cassava starch, roots, peels, pulp, dough, and stalks. The ease of accessibility to these sources is due to the fact that cassava is able to be grown on marginal land or degraded grasslands [275], and to its low price, and it is considered one of the best sources for obtaining bioethanol [276,277].

## 9. Conclusions

Sustainable and biodegradable materials are the hope for the next generation. The devastating environmental problems caused by plastics could be reduced significantly by widely utilizing the reinforcement of biopolymers and natural fibers in composite materials. Development of these renewable composites will improve the atmosphere, reduce the recycling of plastic waste, and decrease petroleum-based materials’ carbon footprint. As potential substitutions for non-biodegradable plastic products, more and more bio-resources are being used for a more sustainable future. Besides that, abundant supply and low-cost features of these green materials have earned them much publicity in recent decades. Nevertheless, the disadvantages related to the use of natural fiber-reinforced biopolymers in composites need to be resolved by further research work. Cassava fiber has excellent tensile strength combined with superior flexural strength, as verified by a number of mechanical tests and research work that enable it to be used in several applications, such as automotive components, auto-industrial, light weight construction, and packaging. Various tree components have been widely used for the production of various local items. Cassava is indeed a promising candidate for strengthening biodegradable polymer composites. Use of cassava in green composites will generally help: (1) minimize the negative environmental effect of synthetic polymers and fibers; (2) decrease the demand for reliance on petroleum products; and (3) grow cassava as a sustainable industrial crop. This will boost rural people’s socio-economic empowerment by raising tax revenues and generating jobs. However, the huge opportunity to use cassava polymer and fiber for various possible industrial applications in the composite industry has not been exploited widely. Nonetheless, in terms of research and testing, more advanced characterizations of cassava fibers and their biocomposites should be carried out. Determining the barrier properties, feasibility, and moisture absorption of cassava biocomposites are essential for successful packaging applications. This is a new area of research and innovation that addresses certain issues that obstruct future industrial applications of cassava fibers, biopolymers, and its composites.

## Figures and Tables

**Figure 1 materials-15-06992-f001:**
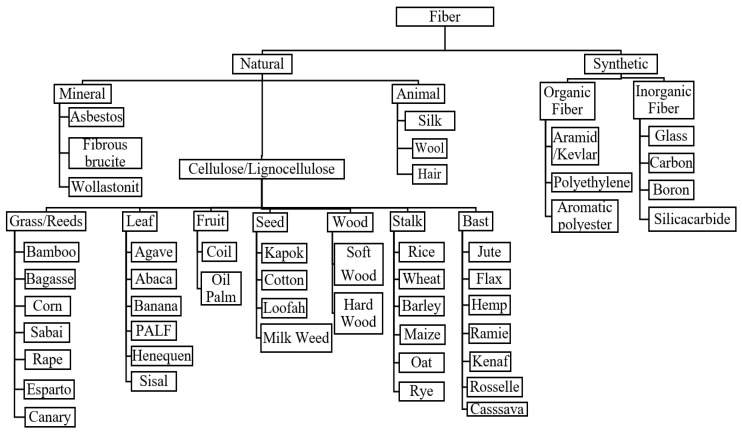
Classification of natural fibers.

**Figure 2 materials-15-06992-f002:**
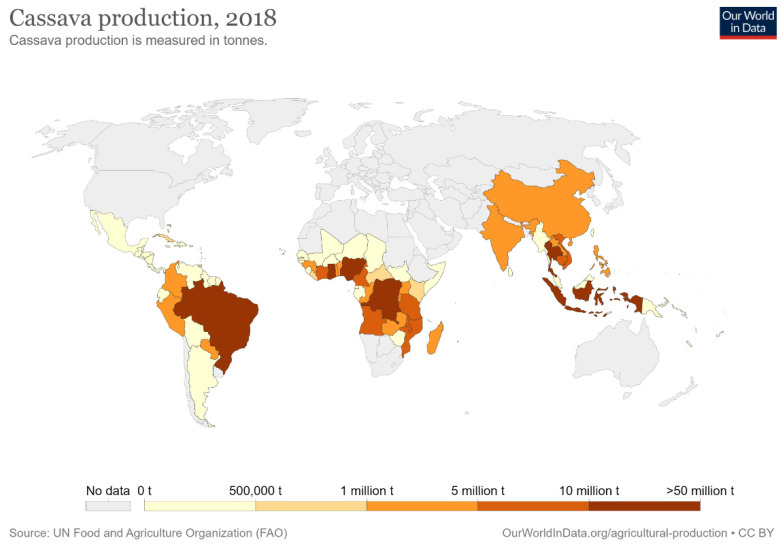
Cassava production statistics by country in 2018 [96].

**Figure 3 materials-15-06992-f003:**
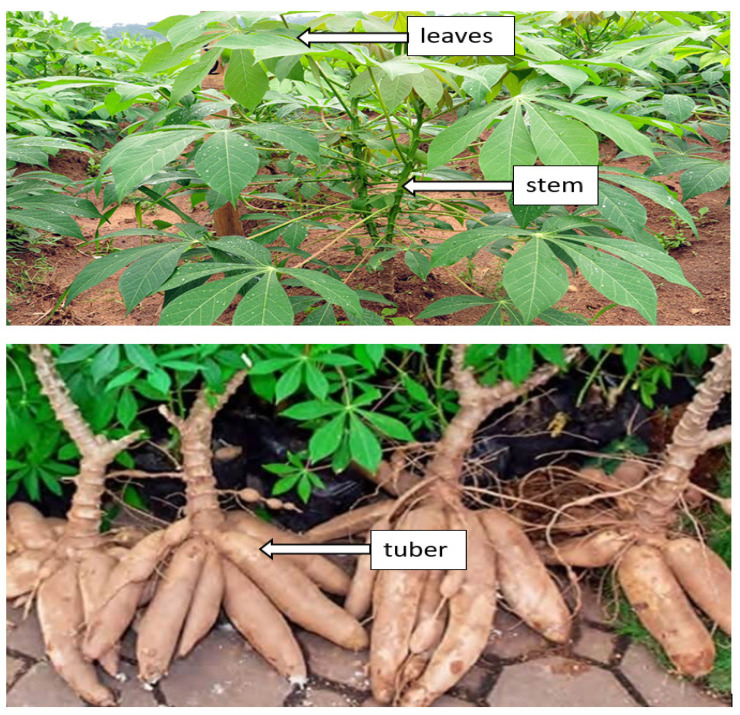
Cassava plant parts (*Manihot esculenta* Crantz). Leaf. This part contains starch and protein, the main building blocks for the cells’ growth and development. Hence, the yields are highly influenced by how stable the leaves are. According to Latif and Müller [98], the major diseases affecting the cassava plant (*Manihot esculenta* Cranz) in Africa are Cassava Brown Streak Disease (CBSD), Cassava Mosaic Disease (CMD), Cassava Bacterial Blight (CBB), and Cassava Green Mite (CGM). Stem. This cassava plant part functions as transport organ by transporting the produced food from leaves to different plant parts for their growth and development. Cassava reproduces via stem cutting; hence, the stem represents a new tool to expand the production of food and fuel materials [99]. Root. Cassava plants are composed of three root types: thick roots, fine white roots, as well as tuberous roots. The thicker roots act as anchors of the plant which grow underground, whereas the tuberous roots collect carbohydrates. Conversely, the fine white roots absorb nutrients and water [100].

**Figure 4 materials-15-06992-f004:**
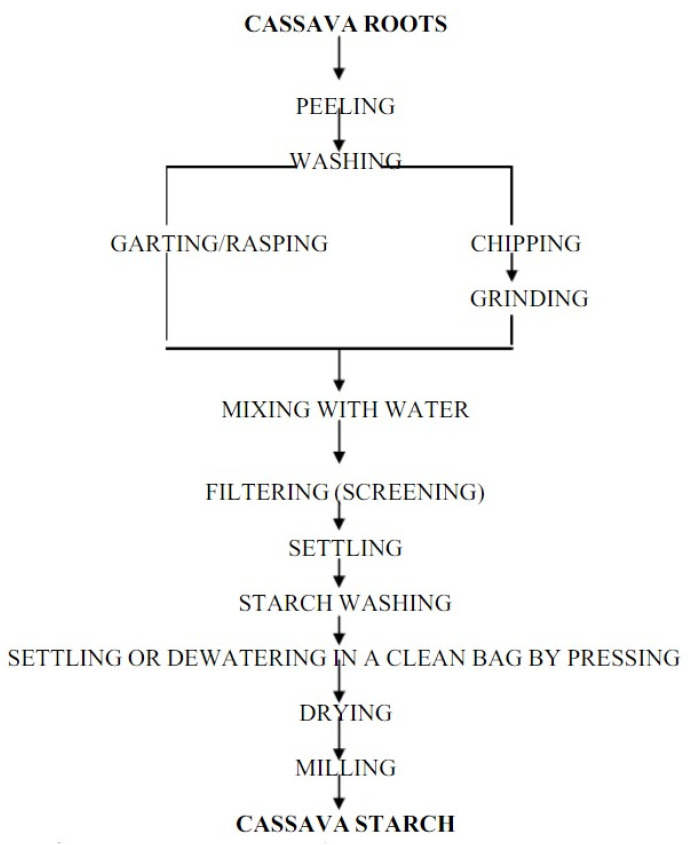
Flow chart for production of cassava starch [119].

**Figure 5 materials-15-06992-f005:**
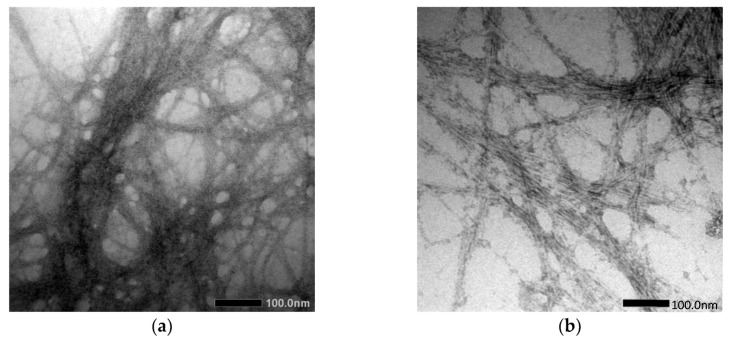
TEM image of (**a**) CNF procedure I and (**b**) CNF procedure II [143].

**Figure 6 materials-15-06992-f006:**
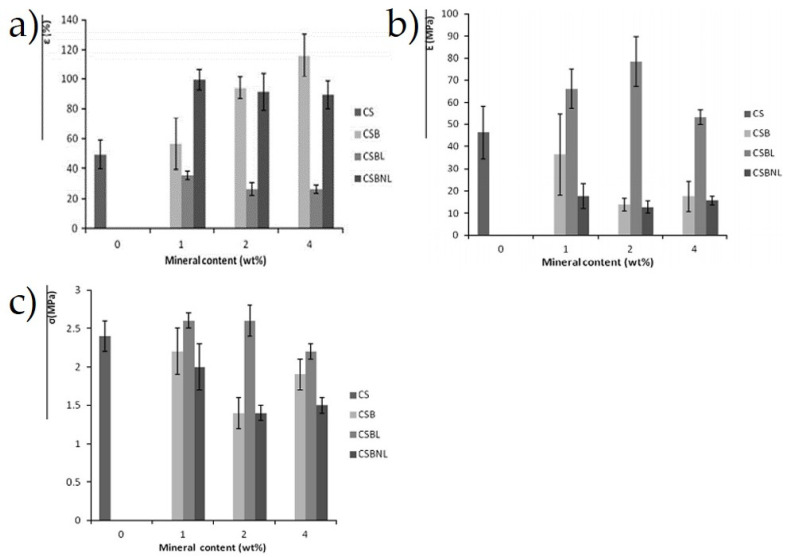
(**a**) Elongation at break (ε), (**b**) tensile strength (σ), and (**c**) Young’s modulus (E) of neat cassava starch film (at 0 wt.%) and 1, 2, and 4 wt.% of CSB-CSBL-CSBNL mineral composite films [151].

**Figure 7 materials-15-06992-f007:**
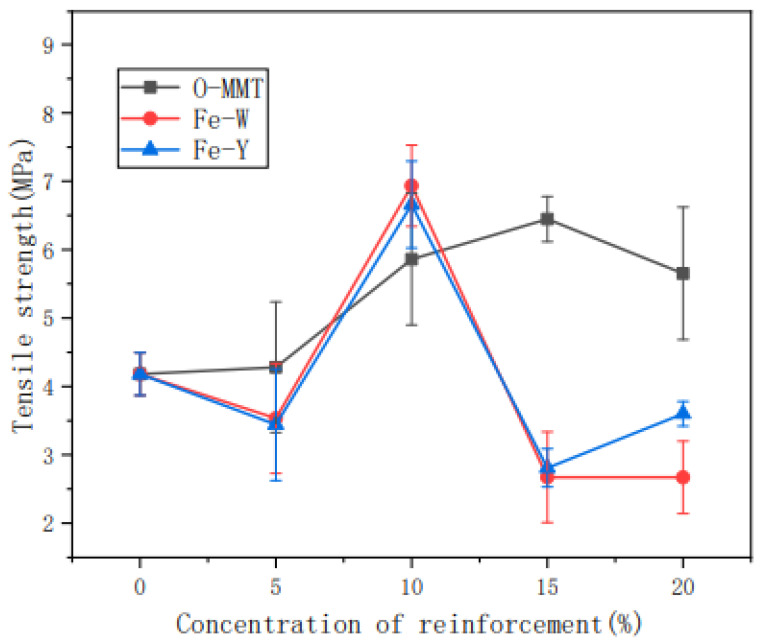
Tensile strength of starch films with different amounts of the reinforcement agent [206].

**Figure 8 materials-15-06992-f008:**
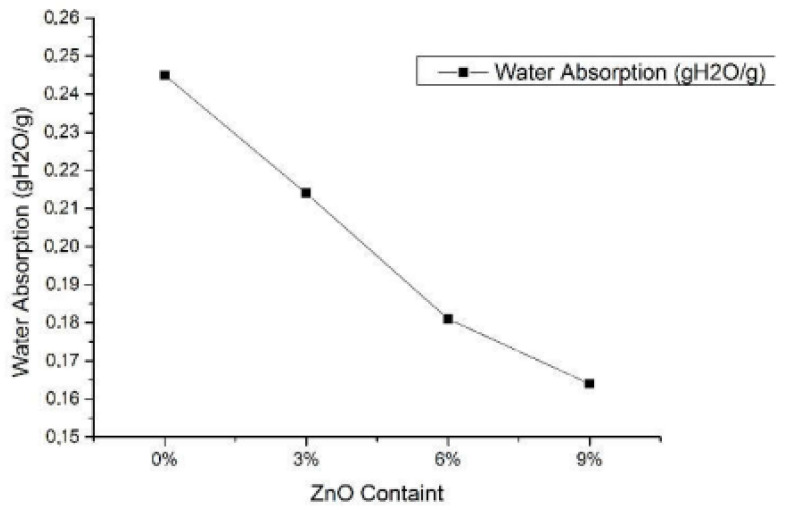
Water absorption result of biofoam with different ZnO addition [173].

**Figure 9 materials-15-06992-f009:**
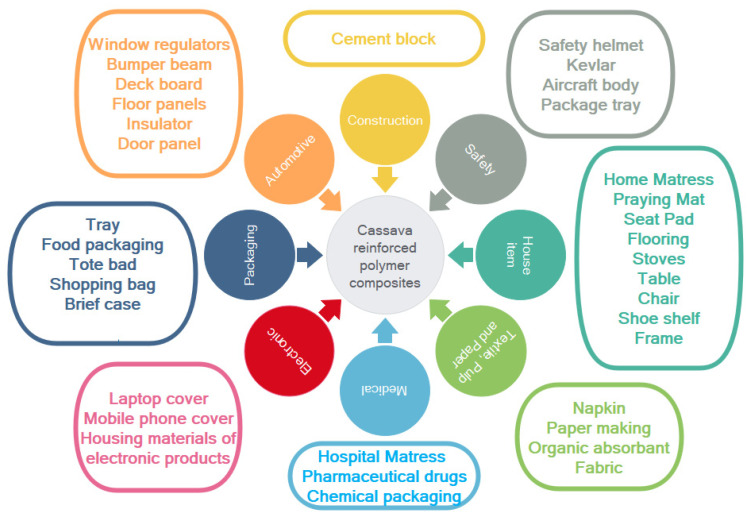
Potential applications for cassava fiber-reinforced polymer composites.

**Figure 10 materials-15-06992-f010:**
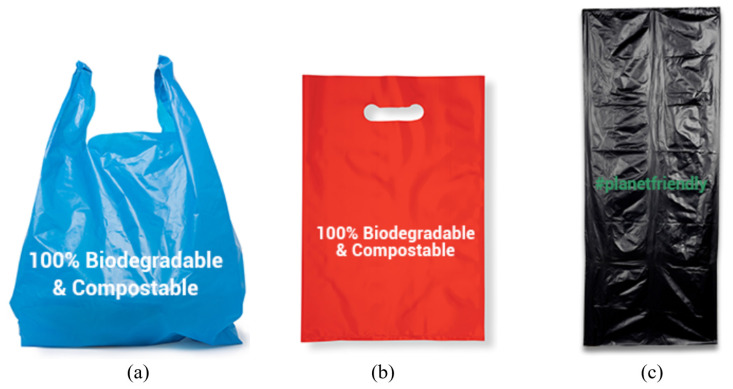
Cassava bags, (**a**) T-shirt bag, (**b**) garbage bag, and (**c**) grip-hole bag.

**Table 1 materials-15-06992-t001:** Pulp content of cassava.

Composition	Content g/100 g Dry Pulp				
	Rattanachomsri et al. [33]	Suwanasri et al. [34]	Kosugi et al. [35]	Sriroth et al. [36]	Virunanon et al. [37]
Starch	60.1 ± 0.1	60.6	68.9 ± 4.0	65.4 ± 4.1	75.1
Crude fiber	n/d	28.1	27.8 ± 0.2	13.2 ± 2.2	n/d
Lignin	2.8 ± 0.06	n/d	n/d	n/d	1.2
Cellulose	15.6	n/d	n/d	n/d	4.1
Protein	n/d	1.8	1.6 ± 0.03	2.1 ± 0.25	n/d
Hemicellulose	4.6	n/d	n/d	n/d	4.2
Fat	n/d	n/d	0.1 ± 0.01	0.2 ± 0.06	n/d

Data are shown as the mean ± 1SD. n/d = no data.

**Table 2 materials-15-06992-t002:** Advantages and disadvantages of natural fibers.

Advantages	Disadvantages
Low specific weight, higher specific strength	Lower impact strength
Renewable, low-energy consumption, low carbon emissions	Sensitive to weather and environment
Facile manufacturing process	Poor moisture resistance, fibers easily swell
Low-cost manufacturing	Restricted maximum processing temperature
Good electrical insulator	Lower durability
Good thermal and acoustic insulating properties	Poor thermal resistance
Biodegradable	Hydrophilic nature, low wetting with hydrophobic polymers

**Table 3 materials-15-06992-t003:** Biochemical compositions of cassava-based residues.

Substrate	Starch (%)	Sugars (%)	Cellulose (%)	Hemicellulose (%)	References
Bagasse	65.6	_	8.1	2.8	[103]
	60.1	_	15.63	4.58	[33]
	56	_	_	_	[104]
	79.45	_	_	_	[105]
	48	_	23	9	[106]
	32.6	18	_	_	[107]
	49.66	_	21.47	12.97	[108]
	57.8	88.4	_	_	[109]
	64	_	_	2.11	[110]
Stem	8.41	_	21.43	11.62	[111]
	_	_	56.4	20.2	[112]
	_	_	35.2	24.3	[113]
	_	_	38.8	7.2	[114]
Peel	41.85	48.22	_	_	[115]
	15.82	_	35.86	9.27	[116]
	67	_	_	_	[117]
Leaf	28.7	29	_	_	[107]
Rhizome	_	_	27.82	39.67	[118]

**Table 4 materials-15-06992-t004:** Physico-chemical compositions of cassava tubers (100 g basis) [1,131].

Composition	Fresh Weight	Dry Weight
Calories	135	335
Moisture (%)	65.5	15.7
Proteins (g)	1.00	1.4
Lipids (g)	0.2	0.5
Starch (g)	32.4	80.6
Fibers (g)	1.1	1.2
Ash (g)	0.9	1.8
Calcium (mg)	26	96
Phosphorus (mg)	32	81
Iron (mg)	0.9	7.9
Sodium (mg)	2	-
Potassium (mg)	394	-
Vitamin B2 (mg)	0.04	0.06
Vitamin C (mg)	34	0
Niacin (mg)	0.6	0.8
Cyanide (%)	-	1.6

**Table 5 materials-15-06992-t005:** Physico-chemical compositions of cassava bagasse (g/100 g dry weight) [1].

Composition	Soccol et al. [133]	Cereda et al. [134]	Pandy et al. [1]	Stertz et al. [135]	Vandenberghe et al. [136]
Moisture	5.02	9.52	5.02–11.2	10.70	11.20
Protein	1.57	0.32	0.32–1.61	1.60	1.61
Lipids	1.06	0.83	0.53–1.06	0.53	0.54
Fibers	50.55	14.88	14.88–50.55	22.20	21.10
Ash	1.10	0.66	0.66–1.50	1.50	1.44
Carbohydrates	40.50	63.85	40.50–63.85	63.40	63.00

**Table 7 materials-15-06992-t007:** Reported work on cassava starch-reinforced natural fibers polymer composites.

Polymer Blend	Fiber	Reference
Native cassava starch, agar (AG), cassava starch (CAS), and arabinoxylan (AX)	Beta zeolite nanocrystal or Na-beidellite	[218]
Cassava flour (CF)/wheat flour (WF)	Cassava stillage residue (CSR)	[158,159]
Sodium cellulose sulfate (NaCS)/cassava starch		[219]
Cassava starch/polylactic acid (PLA)	Cassava bagasse	[220]
Cassava starch/low-density polyethylene	Cotton fibers	[221]
Cassava starch/polyvinyl alcohol (PVA)	Bamboo nanofibrils	[222]
Cassava starch/poly(lactic acid)	Coir fiber	[217]
Cassava starch/polyvinyl alcohol	Sugarcane bagasse fiber	[223]
Cassava starch/poly(butylene succinate)	Maleated poly(butylene succinate)	[224]
Cassava starch/polyvinyl alcohol	Bacterial cellulose fiber	[225]
Cassava/corn starch	Passion fruit peel	[226]
Cassava starch/low-density polyethylene	Cotton fibers	[161]
Starch/poly(vinyl alcohol)	Bamboo cellulose nanofiber	[227]
Cassava/poly(vinyl alcohol) (PVA)	Oil palm empty fruit bunches (OPEFBs) nanocellulose	[228]
Cassava solid waste/bagasse starch (BS)	Bamboo cellulose microfiber (MFC)/epoxidized waste cooking oil (EWCO)	[229]
Cassava/banana starch	Banana fiber	[230]
Cassava Starch/Polybutylene Adipate Terephthalate (PBAT)	-	[231]
Cassava starch/carboxylated styrene-butadiene rubber	Cellulose fiber	[232]
Cassava starch/polylactic acid	-	[233]
Cassava starch/Chitosan	Kraft fiber	[148]

**Table 8 materials-15-06992-t008:** Reported work on cassava starch hybrid polymer composites.

Polymer	Polymer Matrix	Reference
Cassava starch	Cassava peel/cassava bagasse	[156]
Cassava starch	Cassava/sugar palm fiber	[71]
Cassava starch	Cassava bagasse lignocellulose nanofibers (LCNF)/nanoclay (Nclay)	[143]
Cassava starch	Cassava bagasse/kraft paper	[149]
Cassava starch	ZnO/bacterial cellulose	[173]
Cassava starch	Carnauba wax/cashew tree gum-based films	[174]
Cassava starch	Concentrated natural rubber latex/cotton fiber	[175]
Cassava starch	Cellulose pulp fibers modified with deposition of silica (SiO_2_) nanoparticles	[176]
Cassava starch	Oregano essential oil/sugarcane bagasse	[182]
Cassava starch	Cellulose fiber/nanoclay	[188]
Cassava starch	Sugarcane Bagasse Fibers/montmorillonite	[189]
Cassava starch	Ramie fibers CNF/nano PCC tapioca starch	[195]
Cassava starch	Nanofiber straw/ZnO	[197]
Cassava starch	Bamboo fiber, lime juice, epoxidized waste cooking oil	[201]
Cassava starch	Carboxymethylcellulose/lactic acid bacteria	[202]
Cassava starch	ZnO nanorods/PVA electrospun mats/rosemary extract	[203]
Cassava starch	Carboxymethylcellulose/turmeric oil	[213]
Cassava starch	Sisal fiber/carnauba wax	[214]

**Table 9 materials-15-06992-t009:** Tensile strength and elongation at break of cassava and potato starch films with different amount of reinforcing agents [188].

Parameters	Starch	Control	Cellulose	Nanoclay	Cellulose-Nanoclay
Tensile strength (MPa)	Potato	12.03	12.85	14.59	17.75
	Cassava	4.68	6.87	5.78	11.72
Elongation (%)	Potato	13.81	2.12	3.50	0.94
	Cassava	4.62	2.09	4.47	1.32

**Table 10 materials-15-06992-t010:** Tensile strength and elongation at break of CS/CMC composite films with various probiotics loading [202].

Samples	Tensile Strength (MPa)	Elongation at Break (%)
CS/CMC	13.29	65.72
CS/CMC/LPL-0.5%	15.69	51.40
CS/CMC/LPL-1%	15.42	50.22
CS/CMC/LPL-1.5%	14.84	47.24
CS/CMC/LPL-2%	12.73	45.81
CS/CMC/PPE-0.5%	13.10	47.69
CS/CMC/PPE-1%	13.04	44.30
CS/CMC/PPE-1.5%	11.21	42.26
CS/CMC/PPE-2%	11.06	38.00

**Table 11 materials-15-06992-t011:** Reported work on cassava fiber-reinforced polymer composites.

Fiber	Polymer Matrix	Reference
Cassava skin	Polyvinyl alcohol (PVA)	[146]
Cassava bagasse	Polyvinyl-alcohol (PVA) incorporated with clove essential oil (CEO) or oregano essential oils (OEO) and cassava	[147]
Cassava stillage residue (CSR)	Poly(vinyl chloride) (PVC)	[160]
Cassava pulp	Polylactic acid and thermoplastic starch	
Cassava skin	Polyvinyl alcohol (PVA)	[146]
Cassava stillage residue (CSR)	Poly(vinyl chloride) (PVC)	[160]
Cassava and ahipa peels and bagasse	Corn starch	[236]
Cassava nanofibers	Poly(lactic acid)	[237]
Cassava bagasse	Cassava starch	[147]
Cellulose cassava bagasse nanofibrils (CBN)	Cassava starch	[152]
Cassava stillage residue (CSR)	Cassava flour (CF)/wheat flour (WF)	[158,159]
	Poly(vinyl chloride) (PVC)	[160]
	Final egg albumen, cassava starch, sunflower oil	[161]
Cassava peel/cassava bagasse	Cassava Starch	[156]
The remaining fibrous residue of cassava starch extraction	Cassava Starch	[162]
Cassava nanofiber	Cassava starch	[143]
Cassava/sugar palm fiber	Cassava starch	[71]
Cassava bagasse cellulose nanofibrils	Cassava starch	[163]
Cassava bagasse lignocellulose nanofibers (LCNF)/nanoclay (Nclay)	Cassava starch	[143]
Cassava bagasse/kraft paper	Cassava starch	[149]
Cassava bagasse	Cassava starch	[31]
Cassava bagasse (CB)	Cassava starch	[178]
Cassava peel (CP)	Cassava starch	[178]
Cassava peel	Cassava starch	[180]
Cassava roots bagasse	Cassava starch	[187]
Cassava nanofibril	Cassava starch	[138]
Cassava bagasse	Cassava starch	[209]
Cassava cellulose nanocrystals	Cassava starch	[211]

**Table 12 materials-15-06992-t012:** Dynamic-mechanical properties of cassava and corn thermoplastic starch and their composites [147].

TPS	Bagasse Fiber (wt.%)	E(30 °C) (MPa)
Cassava	0	21.5
	5	39.6
	10	95.9
	15	128.5
Corn	0	18.2
	5	59.1
	10	73.0
	15	97.4

## Data Availability

Not applicable.

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
