# Peer review of "Recent Developments in Cassava (Manihot esculenta) Based Biocomposites and Their Potential Industrial Applications: A Comprehensive Review"

_materials, 2022, doi:10.3390/ma15196992_

Round 1

Reviewer 1 Report

The manuscript highlights the importance of the Cassava plant and its fibers or starch-based composites, with some interesting applications. The authors have done a good job in collecting the information; however, further refinement is needed and has been suggested in the following comments to make the content more interesting to read, understand and correlate. 

Review Comments

1. Introduction, Lines 62-65: Please put adequate references to support the properties listed. Also, what is meant by these wastes having “low energy consumption”?

2. What is the advantage of Cassava over other lignocellulose-based sources of starch? It will be nice to have a detailed, in-depth para on this question to highlight the relevance and importance of cassava as a source of starch.

3. Authors are recommended to rearrange the information in the manuscript to highlight the pros and cons for each topic under discussion separately. To say section 2. Natural fibers: Content is unevenly arranged, with advantages and disadvantages crossing.

4. Since this is a review article, including tables with exact or approximate values (with references) of mechanical properties, for example, mechanical strength, biodegradability time frame, density, etc., will be helpful for natural fibers, cassava fibers, and/or starch.  Writing generalized statements for properties does not help much to bring out the required differentiation.

5. Lines 139-175: Summary as a table of advantages/properties listed will be helpful

6. Table 5: Italicize the names (Genus and Species).

7. Table 5: Heading: Physico-chemical properties? Does not match with the content contained therein. Please review the heading

8. Table 5: Is citric acid a product produced using cassava as the substrate? If yes, then in the follow-up column, not sure if a mushroom is a product. Or is biotransformation the product? Please clarify and change the headings/titles of the columns. Also, authors are advised to add more details to this table, such as a) kind of pretreatment method used to break/depolymerize Cassava or mention if the raw form of Cassava was used as the substrate.

9. Section 6: Authors have done a good job in collecting the relevant information. However, this section could have been written with some agenda in mind (reason how cassava as a blend helps the respective research group, and this information could have been represented: as in the example: how adding cassava improved the properties of the other material being used (properties dependent blending, e.g., If need to improve the mechanical strength of cassava fibers, then what materials were used to blend).

10. Table 6: Not sure what is being tried to portray here. More information columns are needed in this table. The same is for the Table following line 617, Table 8.

11. Line 859: What is meant by “Some of their drawbacks can be improved in further studies.” Do the references in these lines reflect how those improvements in properties can be or have been made? It would be nice if the authors could explain what these references refer to in the text in the manuscript itself.

12. Section 8.1: Highlights “blends or use of natural fiber” than cassava fibers. Please clarify if the case study(ies) in this section pertains to cassava fibers or natural fibers in general.

13. Line 127: Replace recycle with Recyclability

14. There are several studies and review articles being published using Cassava starch or its fibers in 2022. These studies have not been referenced in the current study. Authors are advised to include and review the latest studies also in the manuscript.

15. Need sentence restructuring. The tenses used in the manuscript need a little refinement. Authors are advised to revise the manuscript to avoid sentences in the present tense. 

Author Response

Dear Reviewer

Thank you for your constructible valuable critics and suggestions for our manuscript.

We have put as an attached file all of the corrections in detail and we answered all of your questions

Thank you for your attention

Emin BAYRAKTAR (Professor)

Corrseponding author

Reviewer 2 Report

The paper seeks to introduce an approach ‘’Recent developments in cassava (Manihot esculenta) based bio-composites and their potential industrial applications: A comprehensive review”. However, the authors should consider improving upon the quality to further highlight and emphasize. 

1. Manually indicate the magnification footers for figure 5 because the b part is not visible.

2. Put space between each variable and its corresponding unit. The percentages units including most of the other units are written without space. Consider correcting all.

3. The introduction needs to be improved by relating to the mechanics of the studied materials and their mechanical characteristics. The references to be included are: 10.3390/polym14132662, 10.1016/j.polymertesting.2017.09.009, 10.1016/j.compstruct.2021.114698.

4. Images in figure 6 are so blurred. Consider reinserting it as a text but not as a snapshot to increase its natural visibility.

5. Figure 7 has its caption text at the top of the image instead of the bottom. Bring it under the image and make the image looks clear.

6. Give a schematic or a flowchart indicating the flow of ideas in this review work.

7. Add a conclusion at the end of each table highlighting the gap and future perspective.

8. What could be done as a research gap in your conclusion that could possibly be an avenue for a new research area for researchers.

Author Response

Dear Reviewer

Thank you for your valuable critics and suggestions.

We have revised and corrected the manuscripts and we have answered all of the point that you have asked.

Please find as an attached file here that we have prepared for you.

Kind Regards

Emin BAYRAKTAR (Professor)

Round 2

Reviewer 1 Report

Authors have addressed my comments and concerns. I have no additional questions for them.